# Male autism spectrum disorder is linked to brain aromatase disruption by prenatal BPA in multimodal investigations and 10HDA ameliorates the related mouse phenotype

Male sex, early life chemical exposure and the brain aromatase enzyme have been implicated in autism spectrum disorder (ASD). In the Barwon Infant Study birth cohort ($n = 1074$), higher prenatal maternal bisphenol A (BPA) levels are associated with higher ASD symptoms at age 2 and diagnosis at age 9 only in males with low aromatase genetic pathway activity scores. Higher prenatal BPA levels are predictive of higher cord blood methylation across the CYP19A1 brain promoter I.f region ($P = 0.009$) and aromatase gene methylation mediates ($P = 0.01$) the link between higher prenatal BPA and brain-derived neurotrophic factor methylation, with independent cohort replication. BPA suppressed aromatase expression in vitro and in vivo. Male mice exposed to mid-gestation BPA or with aromatase knockout have ASD-like behaviors with structural and functional brain changes. 10-hydroxy-2-decenoic acid (10HDA), an estrogenic fatty acid alleviated these features and reversed detrimental neurodevelopmental gene expression. Here we demonstrate that prenatal BPA exposure is associated with impaired brain aromatase function and ASD-related behaviors and brain abnormalities in males that may be reversible through postnatal 10HDA intervention.

Autism spectrum disorder (ASD or autism) is a clinically diagnosed neurodevelopmental condition in which an individual has impaired social communication and interaction, as well as restricted, repetitive behavior patterns[1]. The estimated prevalence of ASD is approximately 1–2% in Western countries[2], with evidence that the incidence of ASD is increasing over time[3]. While increased incidence is partly attributable to greater awareness of ASD[4], other factors including early life environment, genes and their interplay are important[5]. Strikingly, up to 80% of individuals diagnosed with ASD are male, suggesting sex-specific neurodevelopment underlies this condition[5].

Brain aromatase, encoded by CYP19A1 and regulated via brain promoter I.f[6–8] converts neural androgens to neural estrogens[9]. During fetal development, aromatase expression within the brain is high in males[10] in the amygdala[11,12]. Notably, androgen disruption is implicated in the extreme male brain theory for ASD[13], and postmortem analysis of male ASD adults show markedly reduced aromatase activity compared to age-matched controls. Furthermore, CYP19A1 aromatase expression was reduced by 38% in the postmortem male ASD prefrontal cortex[14], as well as by 52% in neuronal cell lines derived from males with ASD[15]. Environmental factors, including exposure to endocrine-disrupting chemicals such as bisphenols, can disrupt brain aromatase function[16–18].

Early life exposure to endocrine-disrupting chemicals, including bisphenols, has separately been proposed to contribute to the temporal increase in ASD prevalence[19]. Exposure to these manufactured chemicals is now widespread through their presence in plastics and epoxy linings in food and drink containers and other packaging products[20]. Although bisphenol A (BPA) has since been replaced by

✉ e-mail: wah.chin.boon@florey.edu.au

other bisphenols such as bisphenol S in BPA-free plastics, all bisphenols are endocrine-disrupting chemicals that can alter steroid signaling and metabolism[21]. Elevated maternal prenatal BPA levels are associated with child neurobehavioral issues[20] including ASD-related symptoms[22,23], with many of these studies reporting sex-specific effects[20,22–24]. Furthermore, studies in rodents have found that prenatal BPA exposure is associated with gene dysregulation in the male hippocampus accompanied by neuronal and cognitive abnormalities in male but not female animals[20,23,24]. One potential explanation is that epigenetic programming by bisphenols increases aromatase gene methylation, leading to its reduced cellular expression[16] and a deficiency in aromatase-dependent estrogen signaling. If such is the case, it is possible that estrogen supplementation, such as with 10-hydroxy-2-decenoic acid (10HDA), a major lipid component of the royal jelly of honeybees, may be relevant as a nutritional intervention for ASD. Indeed, 10HDA is known to influence homeostasis through its intracellular effects on estrogen responsive elements that regulate downstream gene expression[25,26], as well as its capacity to influence neurogenesis in vitro[27].

Here, we have investigated whether higher prenatal BPA exposure leads to an elevated risk of ASD in males and explore aromatase as a potential underlying mechanism. We demonstrate in a preclinical (mouse) model that postnatal administration of 10HDA, an estrogenic fatty acid, can ameliorate ASD-like phenotypes in young mice prenatally exposed to BPA.

## Human studies

We examined the interplay between prenatal BPA, aromatase function and sex in relation to human ASD symptoms and diagnosis in the Barwon Infant Study (BIS) birth cohort[28]. By the BIS cohort health review at 7-11 years (mean = 9.05, SD = 0.74; hereafter referred to as occurring at 9 years), 43 children had a pediatrician- or psychiatrist-confirmed diagnosis of ASD against the Diagnostic and Statistical Manual of Mental Disorders, Fifth Edition (DSM-5) criteria, as of the 30th of June 2023. ASD diagnosis was over-represented in boys with a 2.1:1 ratio at 9 years (29 boys and 14 girls; Supplementary Table 1). In BIS, the DSM-5 oriented autism spectrum problems (ASP) scale of the Child Behavior Checklist (CBCL) at age 2 years[29] predicted diagnosed autism strongly at age 4 and moderately at age 9 in receiver operating characteristic (ROC) curve analyses; area under the curve (AuC) of 0.92 (95% CI 0.82, 1.00)[30] and 0.70 (95% CI 0.60, 0.80), respectively. The median CBCL ASP score in ASD cases and non-cases at 9 years was 51 (IQR = 50, 58) and 50 (IQR = 50, 51), respectively. Only ASD cases with a pediatrician-confirmed diagnosis of ASD against the DSM-5, as verified by the 30th of June 2023, were included in this report. We thus examined both outcomes (ASP scale and ASD diagnosis) as indicators of ASD over the life course from ages 2 to 9 years (Supplementary Table 1). Quality control information for the measurement of BPA is presented in Supplementary Table 2.

### BPA effects on ASD symptoms at age 2 years are most evident in boys genetically predisposed to low aromatase enzyme activity
Of the 676 infants with CBCL data in the cohort sample, 249 (36.8%) had an ASP score above the median based on CBCL normative data (Supplementary Table 1). From a whole genome SNP array (Supplementary Methods), a *CYP19A1* genetic score for aromatase enzyme activity was developed based on five single nucleotide polymorphisms (SNPs; rs12148604, rs4441215, rs11632903, rs752760, rs2445768) associated with lower estrogen levels[31]. Among 595 children with prenatal BPA and CBCL data, those in the top quartile of the genetic predisposition score, that is, children with three or more variants associated with lower levels of estrogens were classified as 'low aromatase activity' with the remaining classified as 'high aromatase activity' (Fig. 1). Regression analyses stratified by this genetic score and child's sex were performed and an association between high prenatal

BPA exposure (top quartile (>2.18 μg/L) and greater ASP scores was only seen in males with low aromatase activity, with a matched OR of 3.56 (95% CI 1.13, 11.22); *P* = 0.03 (Supplementary Table 4). These findings were minimally altered following adjustment for additional potential confounders. Among males with low aromatase activity, the fraction with higher than median ASP scores attributable to high BPA exposure (the population attributable fraction) was 11.9% (95% CI 4.3%, 19.0%). These results indicate a link between low aromatase function and elevated ASP scores. A sensitivity analysis using an independent weighted *CYP19A1* genetic score confirmed these findings. For the additional score, the Genotype-Tissue Expression (GTEx) portal was first used to identify the top five expression quantitative trait loci (eQTLs; rs7169770, rs1065778, rs28757202, rs12917091, rs3784307) for *CYP19A1* in any tissue type that showed a consistent effect direction in brain tissue. A functional genetic score was then computed for each BIS participant by summing the number of aromatase-promoting alleles they carry across the five eQTLs, weighted by their normalized effect size (NES) in amygdala tissue. This score captures genetic contribution to cross-tissue aromatase activity with a weighting towards the amygdala, a focus in our animal studies. The score was then reversed so that higher values indicate lower aromatase activity and children in the top quartile were classified as 'low aromatase activity' with the remaining classified as 'high aromatase activity'. Again, a positive association between prenatal BPA exposure and ASP scores was only seen in males with low aromatase activity, with a matched OR of 3.74 (95% CI 1.12, 12.50); *P* = 0.03. Additional adjustment for individual potential confounders provided matched ORs between 3.13 to 3.85 (Supplementary Table 5).

### BPA effects on ASD diagnosis at 9 years are most evident in boys genetically predisposed to low aromatase enzyme activity
In subgroup analyses where we stratified by child's sex and unweighted *CYP19A1* genetic score, the results were consistent with those found at 2 years. A positive association between high prenatal BPA exposure and ASD diagnosis was only seen in males with low aromatase activity, with a matched OR of 6.24 (95% CI 1.02, 38.26); *P* = 0.05 (Supplementary Table 4). In this subgroup, the fraction of ASD cases attributable to high BPA exposure (the population attributable fraction) was 12.6% (95% CI 5.8%, 19.0%). In a sensitivity analysis where the weighted *CYP19A1* genetic score was used, a similar effect size was observed in this subgroup; matched OR = 6.06 (95% CI 0.93, 39.43), *P* = 0.06 (Supplementary Table 4).

### Higher prenatal BPA exposure predicts higher methylation of the CYP19A1 brain promoter PI.f in human cord blood
We investigated the link between BPA and aromatase further by evaluating epigenetic regulation of the aromatase gene at birth in the same BIS cohort. *CYP19A1* (in humans; *Cyp19a1* in the mouse) has eleven tissue-specific untranslated first exons under the regulation of tissue-specific promoters. The brain-specific promoters are PI.f[6–8] and PII[17]. For a window positioned directly over the primary brain promoter PI.f, higher BPA was positively associated with average methylation, mean increase = 0.05% (95% CI 0.01%, 0.09%); *P* = 0.009 (Fig. 2). Higher BPA levels predicted methylation across both PI.f and PII as a composite, mean increase per log 2 increase = 0.06% (95% CI 0.01%, 0.10%); *P* = 0.009. Methylation of a control window, comprising the remaining upstream region of the *CYP19A1* promoter and excluding both PI.f and PII brain promoters, did not associate with BPA, *P* = 0.12. These findings persisted after adjustment for the *CYP19A1* genetic score for aromatase enzyme activity. Thus, higher prenatal BPA exposure was associated with increased methylation of brain-specific promoters in *CYP19A1*. Sex-specific differences were not observed. While these effects were identified in cord blood, methylation of *CYP19A1* shows striking concordance between blood and brain tissue (Spearman's rank correlation across the whole gene: ρ = 0.74 (95% CI

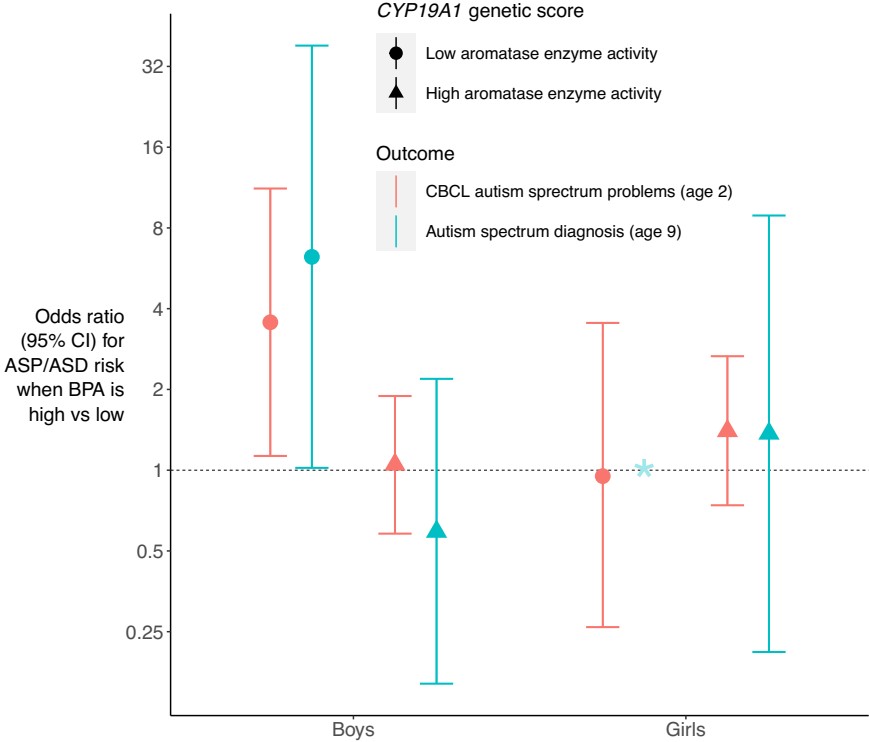

**Fig. 1 | Association between prenatal bisphenol A (BPA) exposure and autism spectrum disorder outcomes stratified by assigned sex at birth and *CYP19A1* genetic score for aromatase enzyme activity.** Conditional logistic regression models were run where participants were matched on ancestry and time of day of urine collection and, for ASD diagnosis at 9 years, each case within these matched groups was individually matched to eight controls based on nearest date of and age at year 9 interview. BPA was classified in quartiles with the top quartile above 2.18 μg/L as high BPA exposure vs the other three quartiles. 'Low aromatase enzyme activity' means being in the top quartile and 'high aromatase enzyme activity' means being in the lower three quartiles of an unweighted sum of the following genotypes associated with lower estrogen levels[31] (participant given 1 if genotype is present, 0 if not): CC of rs12148604, GG of rs4441215, CC of rs11632903, CC of rs752760, AA of rs2445768. 'Greater ASD symptoms' represents a T-score above 50 (that is, above median based on normative data) on the DSM-5-oriented autism spectrum problems scale of the Child Behavior Checklist for Ages 1.5-5 (CBCL). Data are OR ± 95% CI. Source data are provided as a Source Data file. * Since there were only two ASD cases at age 9 in the girls with low aromatase enzyme activity group, the regression model was not run.

0.59, 0.84); over promoter P1.f window: ρ = 0.94 (95% CI 0.54, 0.99)[32]. Thus, prenatal BPA exposure significantly associates with disruption of the *CYP19A1* brain promoter and hence likely the level of its protein product, aromatase.

## Replication of the association between higher BPA levels and hypermethylation of the CYP19A1 brain promoter

Previously, the Columbia Centre for Children's Health Study-Mothers and Newborns (CCCEH-MN) cohort (Supplementary Table 3) found BPA increased methylation of the *BDNF* CREB-binding region of promoter IV both in rodent blood and brain tissue at P28 and in infant cord blood in the CCCEH-MN cohort[33]. In rodents, BDNF hypermethylation occurred concomitantly with reduced BDNF expression in the brain[33]. Re-examining the CCCEH-MN cohort, BPA level was also associated with hypermethylation of the aromatase brain promoter P1.f (adjusted mean increase 0.0040, *P* = 0.0089), replicating the BIS cohort finding.

## Molecular mediation of higher BPA levels and hypermethylation of BDNF through higher methylation of CYP19A1

In BIS, we aimed to reproduce these *BDNF* findings and extend them to investigate aromatase methylation as a potential mediator of the BPA-*BDNF* relationship. A link between aromatase and methylation of the *BDNF* CREB-binding region is plausible given that estrogen (produced by aromatase) is known to elevate brain expression of CREB[34,35]. In BIS, male infants exposed to BPA (categorized as greater than 4 μg/L vs. rest, following the CCCEH-MN study) had greater methylation of the *BDNF* CREB-binding site (adjusted mean increase = 0.0027, *P* = 0.02).

This was also evident overall (adjusted mean increase = 0.0023, *P* = 0.006), but not for females alone (adjusted mean increase = 0.0019, *P* = 0.13). We then assessed whether methylation of aromatase promoter P1.f mediates this association. In both cohorts, aromatase methylation was positively associated with *BDNF* CREB-binding-site methylation in males (BIS, adjusted mean increase = 0.07, *P* = 0.0008; CCCEH-MN, adjusted mean increase = 0.91, *P* = 0.0016). In the two overall cohorts, there was evidence that the effect of increased BPA on *BDNF* hypermethylation was mediated partly through higher aromatase methylation (BIS, indirect effect, *P* = 0.012; CCCEH-MN, indirect effect, *P* = 0.012).

## Prenatal programming laboratory studies—BPA effects on cellular aromatase expression in vitro, neuronal development as well as behavioral phenotype in mice

### BPA reduces aromatase expression in human neuroblastoma SH-SY5Y cell cultures

To validate the findings of our human observational studies on BPA and aromatase expression, we began by studying the effects of BPA exposure on aromatase expression in the human neuroblastoma cell line SH-SY5Y (Fig. 3A). Indeed, the aromatase protein levels more than halved in the presence of BPA 50 μg/L (*P* = 0.01; Fig. 3B) by Western Blot analysis.

### The effects of prenatal BPA exposure on aromatase-expressing neurons within the amygdala of male mice

There is a prominent expression of aromatase within cells of the male medial amygdala (MeA)[11]. To visualize aromatase-expressing cells, we

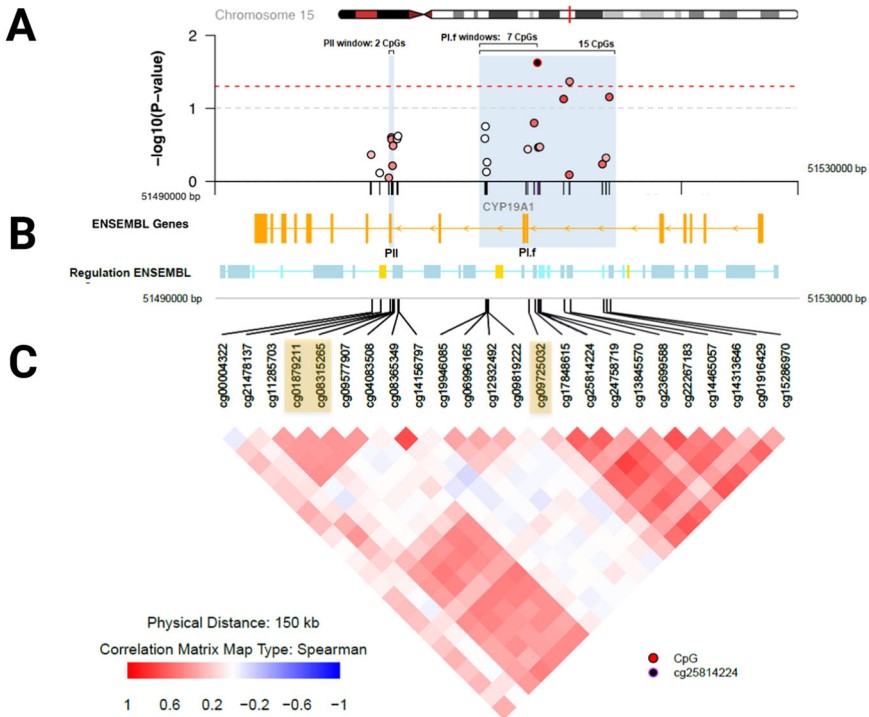

**Fig. 2 | Prenatal BPA-associated aromatase *CYP19A1* gene methylation in cord blood.** Visualized using the coMET R package. **A** Association of individual CpGs along the region of interest with BPA exposure, overlaid with three methylation windows: a 2 CpG window positioned directly on promoter PII, and 7 and 15 CpG windows overlapping PI.f. The red shading reflects each CpG's level of methylation (beta value). **B** The *CYP19A1* gene, running right to left along chromosome 15, and the positions of both brain promoters. Orange boxes indicate exons. **C** A correlation matrix for all CpGs in this region. Highlighted in tan are the two CpGs located within the PII promoter sequence and the single CpG located within PI.f. For the 7 CpG window over promoter PI.f, higher BPA associated positively with methylation, mean increase = 0.05% (95% CI 0.01%, 0.09%); *P* = 0.009,

after adjustment for relevant covariates including cell composition. The BPA-associated higher methylation of the brain promoter PI.f region remained evident when the window was expanded to 15 CpGs (mean increase = 0.06%, 95% CI [0.01%, 0.11%], *P* = 0.04). For PII, the BPA-associated mean methylation increase was 0.07%, 95% CI [-0.02%, 0.16%], *P* = 0.11). BPA also associated positively with methylation across both PI.f and PII as a composite, mean increase = 0.06% (95% CI 0.01%, 0.10%); *P* = 0.009. For the remainder of *CYP19A1*, excluding both PI.f and PII brain promoters, there was no significant association, *P* = 0.12. Higher *CYP19A1* brain promoter methylation leads to reduced transcription[17]. All statistical tests are two sided. Source data are provided as a Source Data file.

studied genetically modified, *Cyp19*-EGFP transgenic mice harboring a single copy of a bacterial artificial chromosome (BAC) encoding the coding sequence for enhanced green fluorescent protein (EGFP) inserted upstream of the ATG start codon for aromatase (*Cyp19a1*)[11] (see Methods). As shown, EGFP (EGFP+) expression in male mice was detected as early as embryonic day (E) 11.5 (Supplementary Fig. 1), indicating that aromatase gene expression is detectable in early CNS development.

To study the effects of prenatal BPA exposure on brain development, pregnant dams were subject to BPA at a dose of 50 µg/kg/day via subcutaneous injection, or a vehicle injection during a mid-gestation window of E10.5 to E14.5, which coincides with amygdala development. This dose matches current USA recommendations[36,37] as well as the Tolerable Daily Intake (TDI) set by the European Food Safety Authority (EFSA) at the time that the mothers in our human cohort were pregnant[28,38]. In these experiments, we observed that prenatal BPA exposure led to a 37% reduction (*P* = 0.004) in EGFP+ neurons in the MeA of male EGFP+ mice compared to control mice (Fig. 3C). These results are consistent with our findings in SH-SY5Y cells that indicate that BPA exposure leads to a marked reduction in the cellular expression of aromatase.

**Prenatal BPA exposure at mid-gestation influences social approach behavior in male mice**

Next, we evaluated post-weaning social approach behavior (post-natal (P) days P21-P24) using a modified three-chamber social interaction test[39] (Fig. 4C). As shown, male mice with prenatal

exposure to BPA were found to spend less time investigating sex- and age-matched stranger mice, when compared with vehicle-treated males (with a mean time ± SEM of 101.2 sec ± 11.47 vs. 177.3 s ± 26.97, *P* = 0.0004; Fig. 4A). Such differences were not observed for female mice prenatally exposed to BPA (Fig. 4A). As a control for these studies, we found that the presence of the EGFP BAC transgene is not relevant to behavioral effects in the test (Supplementary Fig. 2), and the proportions of EGFP transgenic mice were not significantly different across BPA-exposed and vehicle-exposed cohorts.

To determine if the effects of prenatal BPA exposure were developmentally restricted, we delivered subcutaneous injections (50 µg/kg/day) of BPA to pregnant dams at early (E0.5–E9.5), mid (E10.5–E14.5), and late (E15.5–E20.5) stages of gestation. From these experiments, we found that while male pups exposed to BPA in mid-gestation developed a social approach deficit, such behavioral impairments were not observed for early or late gestation BPA exposure (Supplementary Fig. 3). In addition, we performed experiments in which BPA was available to dams by voluntary, oral administration (50 µg/kg/day) during mid-gestation. As shown, a social approach deficit was again observed in male mice (Supplementary Fig. 4), consistent with results from prenatal (mid-gestation) BPA exposure by subcutaneous injections. Thus, we find that prenatal BPA exposure at mid-gestation (E10.5-E14.5) in mice leads to reduced social approach behavior in male, but not female offspring. Notably, the amygdala of embryonic mice undergoes significant development during mid-gestation[40].

## Aromatase knockout (ArKO) male mice have reduced social behavior

Having demonstrated that prenatal BPA exposure reduces aromatase expression in SH-SY5Y cells and affects the postnatal behavior of mice, we next asked if the aromatase gene (*Cyp19a1*) is central to these phenotypes. To address this, we performed social approach behavioral testing (Supplementary Fig. 5) on aromatase knockout (ArKO) mice[41] which have undetectable aromatase expression. The social preference towards the stranger interaction zone compared to the empty zone was only evident for the wildtype ($P = 0.003$ Fig. 4B) but not the ArKO ($P = 0.45$ Fig. 4B). This male-specific social interaction deficit is similar to the BPA exposed pups. Further, postnatal estrogen replacement could reverse the ArKO reduction in sociability seen in males ($P = 0.03$ Supplementary Fig. 5) resulting in a similar stranger-to-empty preference in the E2-treated ArKO as observed for wildtype. The female ArKO pups did not have a sociability deficit (Supplementary Fig. 5).

Further, we did not observe any behavioral differences between ArKO vs WT (or BPA exposed vs unexposed) mice of both sexes in Y-maze test. All groups were able to distinguish the novel arm from the familiar arm. All groups spent significantly more time in the novel arm compared to the familiar arm (Supplementary Fig. 6), excluding major short-term memory, motor and sensory intergroup difference contributions.

## Prenatal exposure to BPA affects repetitive behavior in male mice

Using the water squirt test, we have previously reported that male ArKO, but not female ArKO mice displayed excessive grooming, a form of repetitive behavior, compared to WT mice[42]. Thus, we conducted the water squirt test on BPA-exposed mice to find that male but not female mice exhibited excessive grooming behavior ($P = 0.048$; Fig. 4D). Thus, male prenatal BPA-exposed mice and ArKO mice, but not females, exhibited such repetitive behaviors compared to control mice.

## The development of the MeA is altered in male ArKO mice as well as in prenatal BPA-exposed male mice

The development and function of the amygdala are highly relevant to human brain development and ASD[43,44]. Notably, the medial amygdala (MeA) is central to emotional processing[45], and this tissue is a significant source of aromatase-expressing neurons. Given that aromatase function in the amygdala is significant for human cognition[46] and behavior[12,47], and that aromatase is highly expressed in the mammalian MeA, as particularly observed in male mice[11], we investigated changes to the structure and function of this brain region. We performed stereology analyses on cresyl violet (Nissl)-stained sections of male MeA, we observed a 13.5% reduction in neuron (defined by morphology, size, and presence of nucleolus) number. Compared to the vehicle-exposed males, BPA-exposed males had significantly reduced total neuron number (mean count of 91,017 ± SEM of 2728 neurons vs 78,750 ± SEM of 3322 neurons, $P = 0.046$; Supplementary Fig. 7).

We further examined the characteristics of cells within this amygdala structure in detail using Golgi staining (Fig. 5A, B). We found that the apical and basal dendrites in the MeA were significantly shorter in male BPA-exposed mice vs. vehicle-treated mice (apical: 29.6% reduction, $P < 0.0001$; basal, $P < 0.0001$). This phenotype was also observed for male ArKO vs. WT mouse brains (apical 35.0% reduction, $P < 0.0001$; basal 31.9% reduction, $P < 0.0001$; Fig. 5A). Dendritic spine densities of apical and basal dendrites of male ArKO mice, as well as male mice exposed to BPA, were also significantly reduced (KO vs WT apical, $P = 0.01$; KO vs WT basal, $P < 0.0001$; BPA treated vs vehicle treated apical $P < 0.0001$; BPA treated vs vehicle treated basal, $P = 0.004$; Fig. 5B). The dendritic lengths (Fig. 5A) and

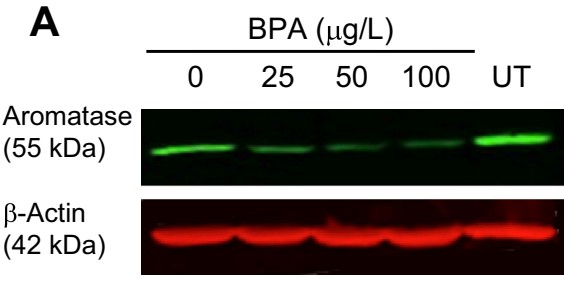

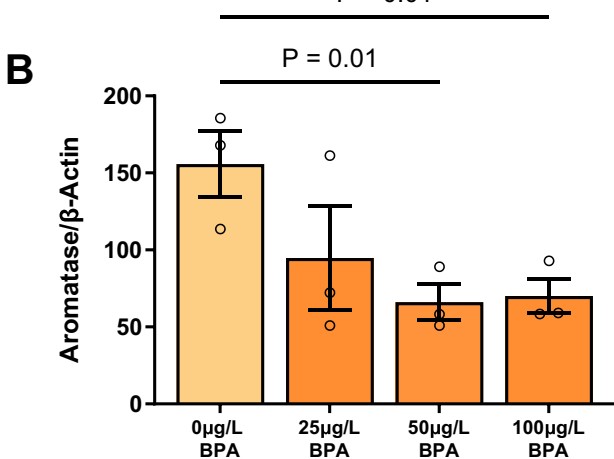

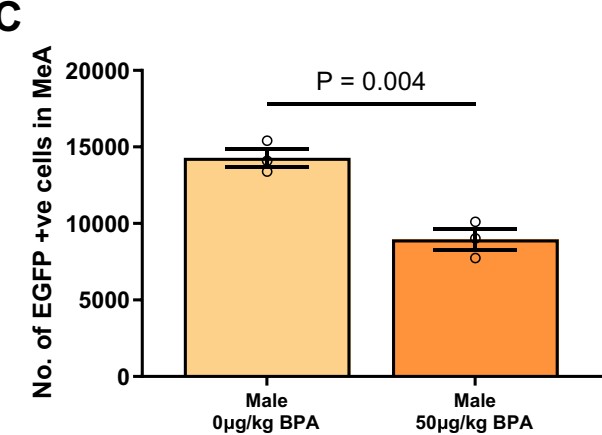

**Fig. 3 | Bisphenol A (BPA) exposure reduces levels of aromatase. A** Western Blot (1 representative blot) demonstrates that increasing BPA concentrations reduced immunoblotted aromatase protein signals (green fluorescence, 55 kDa) in lysates from human-derived neuroblastoma SH-SY5Y cells. Each sample was normalized to its internal house keeping protein β-Actin (red fluorescence, 42 kDa). **B** Aromatase immunoblotted signals in SH-SY5Y cells treated with vehicle or BPA ($n = 3$ independent experiments/group). Five-day BPA treatment of SH-SY5Y cells leads to a significant reduction in aromatase following 50 mg/L(MD = 89, t(6) = 4.0, $P = 0.01$) and 100 mg/L (MD = 85, t(6) = 3.9, $P = 0.01$) BPA treatment, compared to vehicle. **C** BPA treatment (50 μg/kg/day) of *Cyp19*-EGFP mice at E10.5-E14.5 results in fewer EGFP+ neurons in the medial amygdala (MD = -5334, t(4) = 5.9, $P = 0.004$) compared to vehicle mice, $n = 3$ mice per treatment. Independent *t*-tests were used and where there were more than two experimental groups (**B**), *P*-values were corrected for multiple comparisons using Holm-Sidak. All statistical tests were two-sided. Plots show mean ± SEM. Source data are in a Source Data file. Note: UT = untreated.

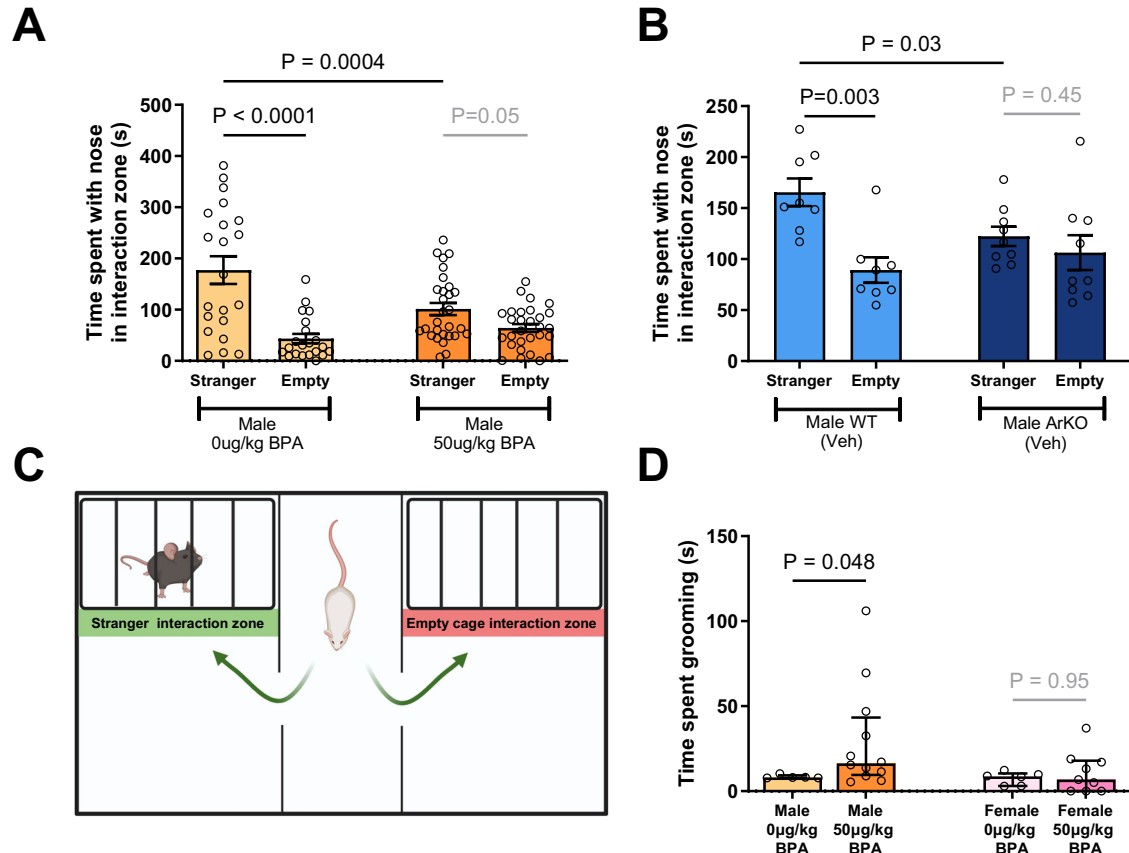

**Fig. 4 | Male BPA-exposed mice and aromatase knockout (ArKO) mice have sociability deficits and increased grooming behavior.** Sociability is the higher proportion of time spent in the stranger interaction zone compared to the empty interaction zone. In the three chamber social interaction test (**A**) BPA-exposed mice ($n = 30$, MD = 75 s, t(96) = 3.7 $P = 0.0004$) spent less time investigating the stranger mouse as compared with male control ($n = 21$) mice. **B** Male ArKO ($n = 8$, MD = 43 s, t(30) = 2.3, $P = 0.03$) mice also spent less time with the stranger compared to male WT littermates ($n = 9$). **C** A schematic of the 3-Chamber Sociability Trial. Created

with BioRender.com. **D** Male BPA-exposed mice ($n = 12$, MD = 8.2, U = 11, $P = 0.048$.) spent more time grooming compared to control ($n = 5$) mice. There were no differences between female BPA-exposed ($n = 9$) and female control ($n = 6$) mice. Independent *t*-tests were used *P*-values were corrected for multiple comparisons using Holm-Sidak. For (**C**), a Mann–Whitney *U* test was used. All statistical tests were two-sided. Plots show mean ± SEM. Source data are in a Source Data file. Note: Veh = vehicle.

spine densities (Fig. 5B) for apical and basal neurites within the MeA of female ArKO mice or BPA-exposed mice were not significantly different compared to control. Thus, in the context of aromatase suppression by prenatal BPA-exposure, or in ArKO mice lacking aromatase, we find that the apical and basal dendrite features within the MeA are affected in a sexually dimorphic manner.

**Prenatal BPA exposure or loss of aromatase in ArKO male mice leads to amygdala hypoactivation and alters behavioral response to a novel social stimulus**

The amygdala, a social processing brain region, is hyporesponsive in ASD (see review ref. [48]). A post-mortem stereology study reported that adolescents and adults diagnosed with ASD feature an ~15% decrease in the numbers of neurons within the amygdala[43]. Also, functional MRI studies report amygdala hypoactivation in participants with ASD compared to controls[49]. Given that the amygdala is a significant source of aromatase-expressing neurons, we next conducted a series of studies to explore how aromatase deficiency influences the male mouse amygdala, using a combination of c-Fos immunohistochemistry, Golgi staining of brain sections, as well as electrophysiological analyses.

To investigate amygdala activation responses after interacting with a stranger mouse, we performed c-Fos immunohistochemistry (a marker for neuronal activation[50]; Supplementary Fig. 8). As shown, prenatal BPA-exposed mice featured 58% fewer c-Fos positive neurons

than in the amygdala of vehicle-exposed mice brains ($P < 0.0001$; Fig. 5C). Similarly, we found that the MeA of ArKO mice had a marked deficit of 67% c-Fos-positive neurons when compared with WT ($P = 0.0002$) mice, which was ameliorated by early postnatal estradiol replacement (Fig. 5C). Therefore, prenatal BPA exposure or loss of aromatase expression in ArKO mice leads to amygdala hypoactivation.

Next, we measured the synaptic excitability (I/O curve) of the MeA using multiple electrode analysis, with excitatory postsynaptic potential (EPSP) output indicative of electrical firing by local neurons. As shown, compared to corresponding controls, we find that MeA excitability (I/O curve) is significantly reduced in male mice prenatally exposed to BPA as well as in male ArKO mice (Figs. 5D and 9D). As shown, at 4-volt input, BPA treatment resulted in a 22.8% lower ($P = 0.02$) excitatory EPSP output than the vehicle treatment, while a 21% reduction ($P = 0.03$) in signal was observed for male ArKO mice compared to male WT mice. Thus, prenatal BPA exposure leads to functional hypoactivation of the amygdala of male mice, and this pattern is also evident in male ArKO mice.

**Prenatal BPA exposure or loss of aromatase in ArKO male mice leads to abnormalities in neuronal cortical layer V as well as brain function**

It has been reported that individuals with ASD show distinct anatomical changes within the somatosensory cortex, including in neurons of cortical layer V[51]. We previously reported that layer V within the

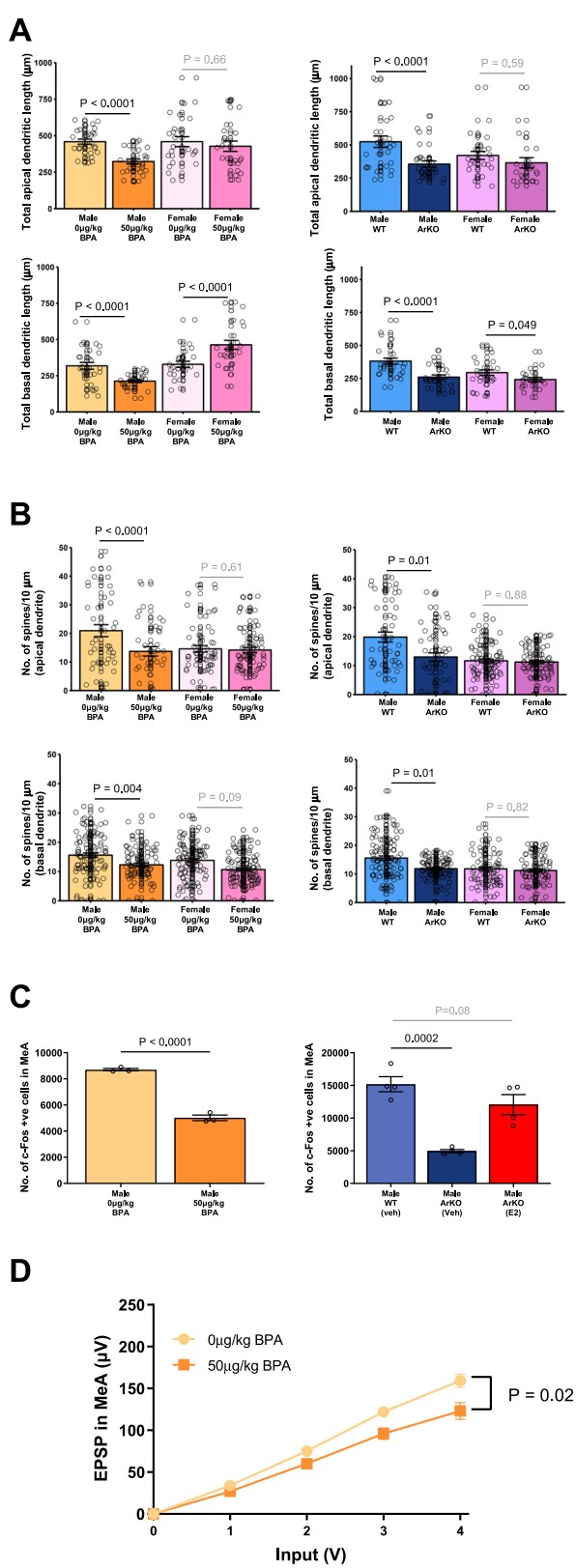

**Fig. 5 | Medial amygdala (MeA) cell alterations in male BPA-exposed and ArKO mice. A** Golgi staining showed shorter apical and basal dendrites in male BPA-exposed (apical: $n = 27$ β = −136μm, 95% CI [−189, −83], $P = 6.0 × 10^{-7}$; basal: $n = 27$ neurons β =−106, 95% CI [−147, −64], $P = 5.1 × 10^{-7}$) and ArKO mice (apical: $n = 27$ β = −194, 95% CI [−258, −130], $P = 2.9 × 10^{-9}$; basal: $n = 27$, β = −121, 95% CI [−143, −100], $P = 1.2 × 10^{-29}$) compared to male vehicle (apical $n = 27$, basal $n = 27$) or WT (apical $n = 27$, basal $n = 29$). Female BPA-exposed mice had longer basal dendrites vs. vehicle ($n = 26$ neurons/group, β = 133, 95% CI [76, 191], $P = 5.5 × 10^{-6}$), while female ArKO mice had shorter basal dendrites vs. WT ($n = 22$ neurons/group, β = −45, 95% CI [−91, −0.2], $P = 0.049$). Significant sex-by-BPA-treatment interaction effects were observed for apical ($P = 0.0002$) and basal ($P = 3.0 × 10^{-11}$) dendritic lengths, and a sex-by-genotype interaction for basal length ($P = 0.003$) but not apical length ($P = 0.19$). **B** Golgi staining showed male BPA-exposed (apical: $n = 39$, β = −7.0, 95% CI [−10.1, −4.0], $P = 5.5 × 10^{-6}$; basal: $n = 90$, β = −3.4, 95% CI [−5.7, −1.1], $P = 0.004$) and ArKO (apical: $n = 51$, β = −6.8, 95% CI [−12.2, −1.3], $P = 0.01$; basal: $n = 97$, β = −3.8, 95% CI [−5.4, −2.2], $P = 5.2 × 10^{-6}$) mice had lower spine densities on apical and basal dendrites vs. vehicle (apical $n = 46$, basal $n = 106$) or WT (apical $n = 53$, basal $n = 109$) mice. Female mice exhibited no spine density differences for BPA exposure (apical $n = 74$, basal $n = 106$) vs. vehicle (apical $n = 61$, basal $n = 103$) and ArKO (apical $n = 94$, basal $n = 86$) vs. WT (apical $n = 88$, basal $n = 83$). There was a significant sex-by-BPA-treatment interaction for apical spine density ($P = 0.0005$) but not basal ($P = 0.99$), and no significant sex-by-genotype interactions (apical: $P = 0.08$; basal: $P = 0.19$). For golgi staining experiments, 3 mice/group with 6–9 neuron measures/mouse. Spine count datapoints represent the number of spines on a single 10μm concentric circle. **C** c-Fos fluorescent immunostaining in adult male PD-MeA revealed fewer c-Fos+ve cells in BPA-exposed ($n = 3$) vs. vehicle mice ($n = 3$; mean difference MD = 3687, $t(4) = 16.12$, $P < 0.0001$) and ArKO ($n = 4$) vs. WT mice ($n = 4$; MD = −10237; $t(4) = 6.48$, $P = 0.0002$). Early postnatal estradiol restored ArKO c-Fos to WT levels ($n = 4$; MD = −3112; t(4) = 1.97, $P = 0.08$). **D** Microelectrode array electrophysiology showed a lower rate of change in EPSP over 1-4 volts in male BPA-exposed mice ($n = 5$ mice, $n = 11$ slices) vs. vehicle ($n = 7$ mice, n = 12 slices; $P = 0.02$). Generalized estimating equations were used clustering by mouse (**A**, **B**) or voltage input (**D**) and assuming an exchangeable correlation structure. For (**C**), independent $t$-tests were used and where there were more than two experimental groups (ArKO analysis), $P$-values were corrected for multiple comparisons using Holm–Sidak. All statistical tests were two-sided. Plots show mean ± SEM. Source data are in a Source Data file.

reductions in dendrites were also reported in male ArKO vs. WT mice (apical $P < 0.0001$; basal $P = 0.02$; Fig. 6A). Furthermore, we found that dendritic spine densities on apical dendrites were also reduced (BPA-exposed mice vs. vehicle, $P = 0.04$; ArKO vs. WT mice, $P = 0.01$ (Fig. 6B).

To explore the effects of reduced aromatase on cortical activity, we performed electrocorticography (ECoG) recordings from mice in both experimental models (Fig. 6C). As shown, spectral analysis revealed an increased power in the range of 4–6 Hz for ArKO vs. WT mice (4 Hz $P = 0.0006$, 5 Hz $P < 0.0001$, 6 Hz $P < 0.0001$; Fig. 6D) and at 8 Hz for BPA-exposed vs. vehicle mice ($P = 0.01$; Fig. 6C). These data indicate that BPA-exposure or loss of aromatase in ArKO mice affects cortical activity, a result which is reminiscent of cortical dysfunction evidenced by EEG recordings on human participants diagnosed with ASD[53].

**Molecular docking simulations indicate 10HDA is acting as a ligand at the same site as BPA on Estrogen Receptors α and β**

It has been reported that BPA interferes with estrogen signaling through its competitive interaction and binding with estrogen receptors α (ERα) and β (ERβ)[54]. To explore this in the context of our findings, we used high-resolution in silico 3D molecular docking simulations to model the binding affinity of the natural ligand 17β-estradiol, the putative ligand BPA, as well as a putative therapeutic ligand of interest 10HDA with ERα (Protein Data Bank (PDB) ID: 5KRI) and ERβ (PDB ID: 1YYE). As shown, our spatial analysis indicated that all three ligands have robust binding affinity (Fig. 7; Supplementary Movie 1). However, while docking alignment revealed that the predicted fit for 10HDA is strikingly similar to that of 17β-estradiol[25,55], BPA

somatosensory cortex is disrupted in ArKO mice[52]. Thus, we performed Golgi staining to study the apical and basal dendrites of neurons within layer V of the somatosensory cortex following prenatal BPA exposure, as well as in ArKO mice. As shown, we found that apical and basal dendrite lengths of layer V cortical neurons were significantly decreased in male mice prenatally exposed to BPA, compared with vehicle-treated mice (apical $P = 0.04$; basal $P < 0.0001$, Fig. 6A). Such

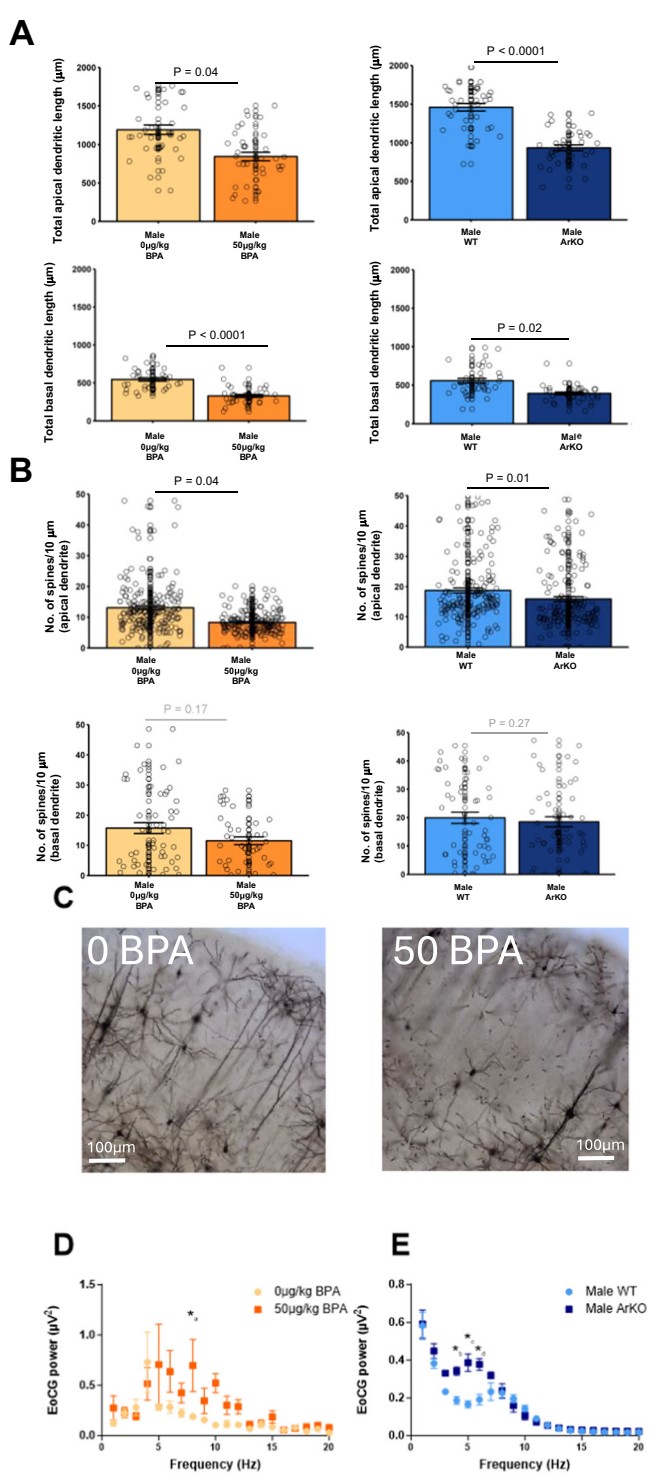

**Fig. 6 | Altered structure and function of cortex in male BPA-exposed and ArKO mice. A** Golgi staining showed shorter apical and basal dendrites in male BPA-exposed (apical: $n = 36$, β =-350μm, 95% CI [−679, −20], $P = 0.04$; basal: $n = 36$, β = −217, 95% CI [−315, −119], $P = 1.4 \times 10^{-5}$) and ArKO mice (apical: $n = 35$, β = −541.9, 95% CI [−666, −417], $P = 1.3 \times 10^{-17}$; basal: $n = 35$, β = −163, 95% CI [−308, −17], $P = 0.02$) compared to male vehicle (apical $n = 35$, basal $n = 36$) or WT(apical $n = 36$, basal $n = 36$). **B** Golgi staining showed male BPA-exposed (apical: $n = 186$, β = −4.7, 95% CI [−9.2, −0.2], $P = 0.04$; basal: $n = 40$ β = −6.7, 95% CI [−16, 2.8], $P = 0.17$) and ArKO (apical: $n = 148$ β = −4.4, 95% CI [−7.7, −1.0], $P = 0.01$; basal: $n = 51$ β = −5.2, 95% CI [−14.4, 4.1], $P = 0.27$) mice had lower spine densities on apical but not on basal dendrites vs. vehicle (apical $n = 189$, basal $n = 56$) or WT mice (apical $n = 185$, basal $n = 55$). For golgi staining experiments, 3 mice/group with 9–12 neuron measures/mouse. Spine count datapoints represents the number of spines on a single 10 μm concentric circle. **C** Representative photomicrographs of golgi stained cortical neurons, scale bar is 100μm. **D** Electrocorticograms (ECoG) revealed an increased in the average spectral power at 8 Hz in BPA-exposed ($n = 4$; *$_a$ 8 Hz MD = −0.5; t(325) = 3.4 $P = 0.01$) mice and (E) 4–6 Hz in ArKO mice ($n = 4$; *$_b$ 4 Hz MD = −0.2; t(120) = 4.3, $P = 0.0006$; *$_c$ 5 Hz MD = −0.2, t(120) = 6.1, $P < 0.0001$; *$_d$ 6 Hz MD = −0.2, t(120) = 5.2 $P < 0.0001$) vs. vehicle ($n = 7$) or WT ($n = 4$)mice. Generalized estimating equations were used clustering by mouse (Panels A, B) and assuming an exchangeable correlation structure. For (**D**) and(E) Independent $t$ test were used used, $P$-values were corrected for multiple comparisons using Holm-Sidak. All statistical tests were two-sided. Plots show mean ± SEM. Source data are in a Source Data file.

Co-administration with 10HDA ameliorated these adverse effects of BPA (Fig. 8A, 10HDA + BPA; quantified in Fig. 8B, C).

### In vivo effects of 10HDA on BPA mouse model
Guided by our findings in cultured neurons, we next investigated the effects of postnatal 10HDA administration on mice prenatally exposed to BPA at mid-gestation, as follows. After weaning, pups (six litters, 3 weeks of age) were administered daily injections of 10HDA (0 and 500 μg/kg/day; dissolved in saline, i.p.) for 3 weeks, following which pups were assessed for behavioral phenotypes. Strikingly, 10HDA treatment significantly improved social interaction (Fig. 9A). To determine whether the effect of 10HDA administration is permanent, all treatments were withdrawn for 3 months, and mouse behaviors were subsequently re-tested. Withdrawal of 10HDA treatment in BPA-exposed male mice resulted in a deficit in social interaction (Fig. 9B), and this deficit was once again ameliorated by a subsequent 10HDA treatment (Fig. 9C) at 5 months of age, in adulthood. Taken together, these data demonstrate that continuous, postnatal 10HDA administration is effective for ameliorating social interaction deficits in male mice following prenatal BPA exposure.

Next, we wanted to determine if hypoactivity arising from the absence of aromatase in the amygdala may be influenced by 10HDA. To address this question, we studied ArKO mice using multiple-electrode analyses, following 3 weeks of treatment with 10HDA (500 μg/kg/day, i.p.). As shown in Fig. 9D, the electrical activity of the male ArKO amygdala treated with 10HDA was similar to male WT activity levels, whereas saline-treated male ArKO amygdala showed significantly lower activity ($P = 0.03$) when stimulated by an input/output paradigm, suggesting that 10HDA treatment was effective to compensate the absence of aromatase. Therefore, we interpret these results to suggest that 10HDA restores signaling deficits arising from aromatase deficiency. Given that prenatal BPA exposure suppresses aromatase, 10HDA supplementation may be relevant to aromatase-dependent signaling in that context as well.

### Transcriptomic studies of the fetal brain cortex and cortical cell cultures
MiSeq Next-Gen Sequencing was performed on the transcriptome libraries generated from the brain cortex of the E16.5 fetuses after maternal mid-gestation BPA or vehicle exposure. The action of 10HDA was analyzed by RNAseq of transcriptome libraries from total RNA

showed a greater mismatch (Fig. 7D), consistent with previous reports that BPA is 1000-fold less estrogenic than the native ligand[56]. Thus, at least for ERα and Erβ, we find that 10HDA may be effective as a competitive ligand that could counteract the effects of BPA on estrogen signaling within cells.

### In vitro effects of BPA and 10HDA in primary fetal cortical cell cultures from male brains
Examining male fetal primary cortical culture, BPA alone shortened neurite lengths (Fig. 8A, BPA; quantified in Fig. 8B-C) and decreased the spine density. BPA treatment reduced both neurite length ($P = 0.0004$) and spine densities ($P < 0.0001$; Fig. 8A, BPA; quantified in Fig. 8B–C).

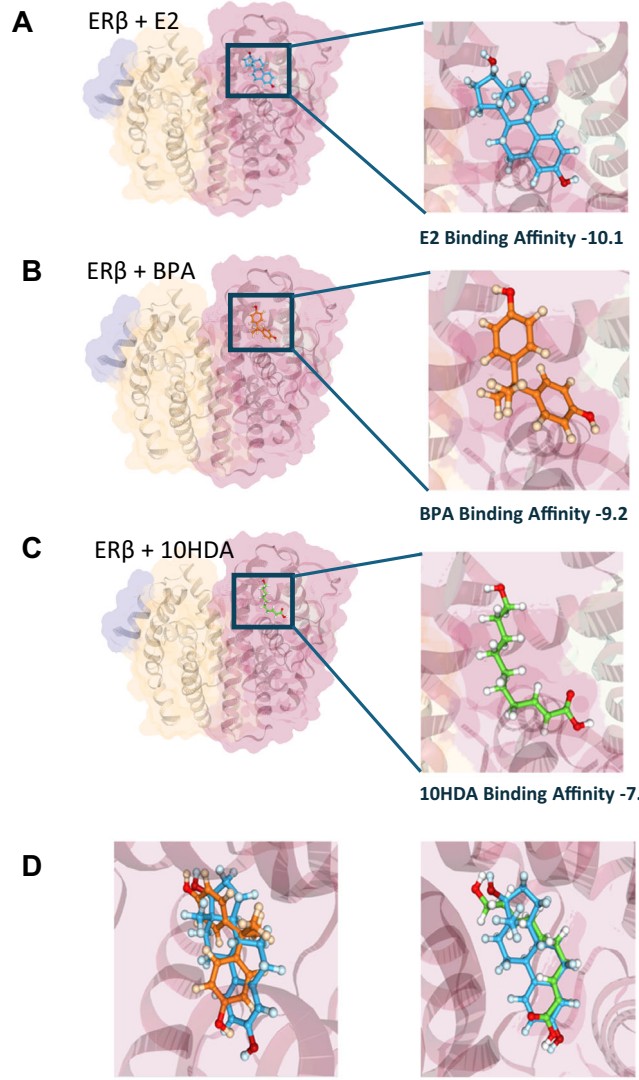

**A** ERβ + E2

**E2 Binding Affinity -10.1**

**B** ERβ + BPA

**BPA Binding Affinity -9.2**

**C** ERβ + 10HDA

**10HDA Binding Affinity -7.9**

**D**

**Fig. 7 | Molecular docking of E2, BPA and 10HDA with estrogen receptor β.** In silico molecular docking analysis of estrogen receptor β (ERβ, Protein Data Bank (PDB) ID: 1YYE; encoded by the ES gene) using the DockThor platform, showing binding predictions for (**A**) the native ligand 17β-estradiol (E2), (**B**) bisphenol A (BPA), (**C**) Trans-10-hydroxy-2-decenoic acid (10HDA), and (**D**) E2 and BPA (left) and E2 and 10HDA (right) superimposed for spatial alignment comparison. While the molecular affinities of BPA and 10HDA for Erβ were comparable (−9.2 vs. −7.9, respectively), 10HDA aligns better with the binding conformation of the endogenous ligand E2, which activates the receptor. BPA is previously reported as sub-optimally estrogenic[106]− >1000-fold less compared to natural estradiol[54,56]−whereas 10HDA has an estrogenic role in nature[25,55]. Thus, 10HDA may compensate for E2 deficiency caused by a reduction in aromatase enzyme, and in competition with binding by BPA. Please see Supplementary Movie 1 for a video of the above molecular docking of ERβ with E2 superimposed with BPA and Supplementary Movie 2 for the above molecular docking of ERβ with E2 superimposed with 10HDA.

extracted from primary mouse fetal cortical cultures treated with vehicle or 10HDA. Firstly, pathway analysis of the RNAseq data was performed for Gene Ontology (GO) categories using the clusterProfileR R package. No individual pathways in the BPA analysis survived correction for multiple comparisons using an agnostic (non-candidate) approach. Further candidate investigation using the binomial test showed a significant inverse effect of BPA and 10HDA on pathways previously linked to autism[57], with 10HDA treatment countering the effects of BPA on these pathways (Supplementary Fig. 9). Based on our Golgi staining experimental findings relating to altered dendrite

morphology, we further assessed the category "dendrite extension" as a candidate pathway. Genes in this pathway were downregulated by BPA (Supplementary Figs. 10A, 9A) and upregulated by 10HDA (Supplementary Figs. 9B, 10B). More broadly, Fisher's exact test showed a significant BPA-associated down-regulation (*P* = 0.01), and 10HDA-associated up-regulation (*P* = 0.0001), of pathways with the terms "axon" and "dendrite". Notably, the majority (82%; 9 of the top 11 available) mid gestational biological processes whose activity is over-represented in induced pluripotent stem cells of autism cases vs. controls[57] were impacted by BPA, and in the opposite direction to 10HDA (*P* = 0.03; Supplementary Fig. 9).

Next, we performed pathway enrichment analysis, also using a candidate pathway approach, of the RNAseq data using Ingenuity (Fig. 10). Strikingly, the effects of BPA and 10HDA on gene expression were diametrically opposed across many functional domains (Fig. 10). For example, the canonical pathways "Synaptogenesis Signaling pathway" and "CREB signaling" were downregulated by BPA but upregulated by 10HDA. Similarly, key brain functions, e.g., growth of neurites and neural development were down regulated by BPA and reciprocally upregulated by 10HDA (Fig. 10). Taken together, prenatal BPA exposure is detrimental to gene expression through a mechanism that may be ameliorated by postnatal 10HDA administration. The full list of differentially expressed genes can be found in Supplementary Dataset 1.

## Discussion

Here, we report that prenatal BPA exposure leads to ASD endophenotypes in males, and that this involves the actions of the aromatase gene, as well as its functions in brain cells. Our multimodal approach, incorporating both human observational studies and preclinical studies with two mouse models, offer significant insight into how prenatal programming by BPA disrupts aromatase signaling to cause anatomical, neurological, as well as behavioral changes reminiscent of ASD in males.

In the Barwon Infant Study (BIS) human birth cohort the adverse effect of high prenatal BPA exposure on ASD symptoms (ASP score) at age 2, and clinical ASD diagnosis at age 9, was particularly evident among males with a low aromatase enzyme activity genetic score. Further, studying cord blood gene methylation as an outcome we have demonstrated that BPA exposure specifically methylated the offspring *CYP19A1* brain promoter in the BIS cohort, replicated in the CCCEH-MN cohort. Previously, in a meta-analysis of two epigenome-wide association studies (EWAS), two CpGs in the region around the brain promoters PI.f and PII were significantly associated (*P* < 0.05) with ASD in both EWAS[58]. Past work[16], and our findings of BPA reduced aromatase expression in a neuronal cell culture and reduced aromatase-eGFP expression in mouse brain, are consistent with the brain-specific suppression of aromatase expression. We also demonstrated that BPA exposure led to a reduction in steady-state levels of aromatase in a neuronal cell line. Further, we replicated past work that higher prenatal BPA levels are associated with *BDNF* hypermethylation[33], previously demonstrated to be associated with lower *BDNF* expression in males[33].

We find that prenatal BPA exposure at mid-gestation in mice induces ASD-like behaviors in male but not female offspring, concomitant with cellular, anatomical, functional, and behavioral changes (Supplementary Fig. 11 and Supplementary Fig. 12). We found that these features were also observed in male ArKO mice, and this is important because aromatase expression is disrupted in BPA-treated male mice. In the Y-maze test, both the ArKO and BPA offspring did not show any differences with the respective control indicating that there are no major memory, sensory or motor issues in these animals. Given the distinct parallels between the effects of BPA that suppresses aromatase, as well as ArKO mice that lack aromatase, we surmise from our studies that BPA disrupts aromatase function to influence the male mouse brain which manifests as: (i) reduced excitatory postsynaptic

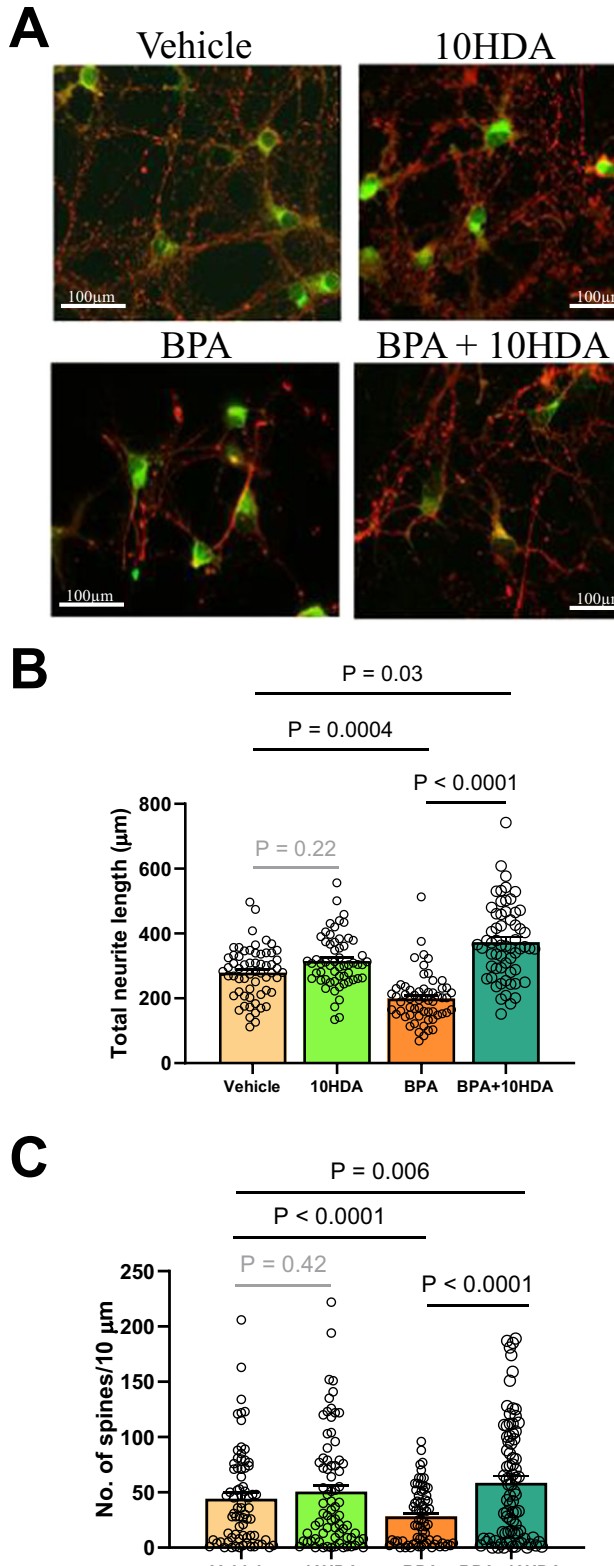

**Fig. 8 | BPA-exposed male cortical neurons and 10HDA exposure in vitro.**
**A** Representative photomicrographs of primary cultures of embryonic (ED15.5) mouse cortical neurons, red staining is βIII tubulin and green is aromatase. Scale bar is 100μm. **B** Compared to the BPA group, the vehicle group (β = 79.9, 95% CI [36, 124], $P = 0.0004$) and BPA + 10HDA group (β = 174, 95% CI [102, 247], $P = 2.4 \times 10^{-7}$) have significantly longer neurites. The BPA + 10HDA group (β = 94, 95% CI [8, 180], $P = 0.03$) has longer neurites compared to the vehicle group, and there is no difference between the vehicle and 10HDA groups. **C** Compared to the BPA group, the vehicle group(β = 16, 95% CI [11, 21], $P = 1.7 \times 10^{-8}$) and BPA + 10HDA group (β = 30, 95% CI [21, 40], $P = 8.4 \times 10^{-10}$) have significantly higher spine densities. The BPA + 10HDA group (β = 14, 95% CI [4, 24], $P = 0.006$) has a higher spine density compared to the vehicle group, and there is no difference between the vehicle and 10HDA groups. $n = 10$ neurons/group. Primary cortical cell culture was obtained from 12 male mouse embryoes. Spine count datapoints represent the number of spines on a single 10μm concentric circle. Generalized estimating equations were used clustering by mouse and assuming an exchangeable correlation structure. All statistical tests were two-sided. Plots show mean ± SEM. Source data are in a Source Data file.

individuals with ASD[51]. Furthermore, we investigated our gene expression dataset and found that the majority of the top biological processes over-represented in cells derived from ASD cases compared to non-cases in a human pluripotent stem cell analysis with a focus on mid gestational brain development[57] were impacted in opposite directions by BPA compared to 10HDA in our gene expression studies on the male mouse brain (Supplementary Fig. 9). Of note, the sexually dimorphic effect we report is consistent with work of others demonstrating that prenatal BPA exposure of rodents led to dysregulation of ASD-related genes with neuronal abnormalities, and learning and memory problems only in males[59].

In our investigations of BPA, we recognized that 10HDA may be a suitable compound as a ligand in the context of brain ER signaling[27] because of its positive effect on gene expression through stimulation of estrogen responsive DNA elements[25] and its role in neurogenesis[27]−characteristics that altogether may compensate for a relative lack of aromatase-generated neural estrogens. Administration of 10HDA alongside BPA protected neuronal cells in culture from the adverse sequelae observed for BPA alone at the same dose. Three weeks of daily postnatal 10HDA treatment significantly enhanced the sociability of the male BPA-exposed mice and dendrite morphology in primary cell culture. The adverse decrease in dendrite lengths and spine densities of the BPA-exposed mice was also corrected by 10HDA administration (Fig. 8). Furthermore, postnatal 10HDA treatment restored amygdala electrical activity in the ArKO mice, indicating that 10HDA likely acts downstream of, rather than directly upon, the aromatase enzyme, given that ArKO mice lack functional aromatase. Transcriptomic analyses revealed that 10HDA upregulated, whereas BPA downregulated, gene expression for fetal programming such as for synaptogenesis and growth of neurites. Some of these pathways could be activated by factors downstream of aromatase, such as 17β-estradiol (Supplementary Fig. 13). In this study, the ArKO model was useful because it provided an estrogen deficient comparison[41]. We were able to demonstrate that early postnatal E2 administration restored both MeA neural activation and social preference behavior in the ArKO males.

The molecular docking simulations indicate that ERα and ERβ both comprise docking sites for 10HDA and BPA, however, 10HDA is strongly estrogenic[25,55] while BPA is greater than 1000-fold less potent than natural estrogen[54]. Such differences in binding are likely relevant to the diverse transcriptomic effects observed in the cells we analyzed by RNAseq.

Strengths of this study include the multimodal approach to test the hypothesis of the interplay of BPA, male sex, and aromatase suppression. In our human epidemiological studies, extensive information was available to allow confounding to be accounted for using matched

potentials in the amygdala, (ii) reduced neuron numbers as well as dendritic lengths and spine densities for neurons within the MeA, (iii) altered cortical activity as recorded by ECoG concomitant with decreased dendritic length and spine density in layer IV/V somatosensory cortical neurons, as well as (iv) enhanced repetitive behaviors and reduced social approach to a stranger. These results in mice are consistent with studies with human participants that report abnormal neuronal structure in these comparative regions within the brains of

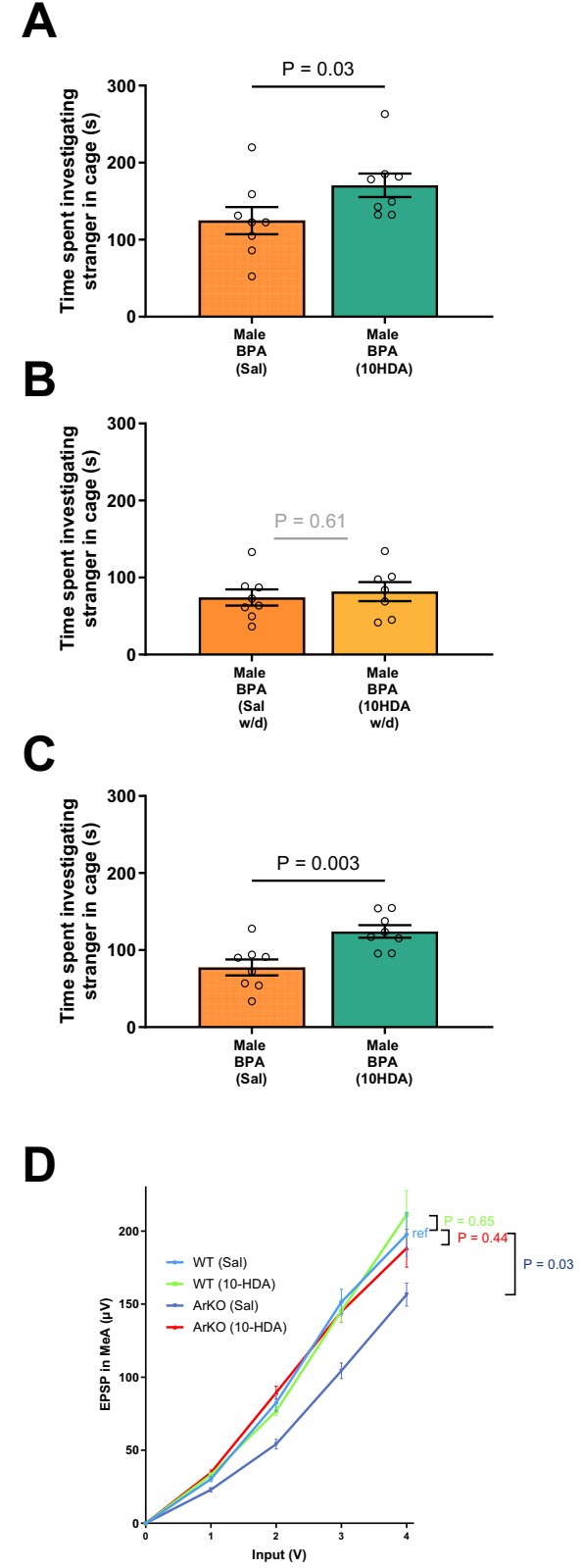

**Fig. 9 | Social approach and excitatory postsynaptic potential deficits are ameliorated by postnatal 10HDA. A** 10HDA treatments increased social approach in males (n = 10/group, MD = 41.14, U = 11, P = 0.03,) but not females (n = 8/group), compared to saline controls. After 3 months of treatment withdrawal (**B**), male mice (saline n = 8, 10HDA n = 7) no longer spent more time interacting with strangers, compared to vehicle treatment. **C** When male mice (n = 8/group) were subsequently treated with a second round of 10HDA, social approach behavior was once again significantly elevated (MD = 39.2, U = 5, P = 0.003), indicative of a rescue of this behavioral effect. **D** Compared to the WT Saline (n = 10) group (β = 57.1, 95% CI [47.4, 66.8]), EPSP increases at a 21% lower rate with increasing input in the ArKO Saline (n = 14) group (β = 45.1 μV, 95% CI [40.1, 50.1], P = 0.03). No differences in slope were detected when comparing the WT Saline group with each of the other two treatment (WT n = 18, KO n = 12) groups. Mann–Whitney U tests were used and for (**D**), Generalized estimating equations were used clustering by voltage input (**D**) and assuming an exchangeable correlation structure. All statistical tests were two-sided. Plots show mean ± SEM. Source data are in a Source Data file. Note: Sal = saline, w/d = withdrawal.

(Supplementary Fig. 11); and the consistency with which our experimental laboratory work maps to prior studies of people with ASD (summarized in Supplementary Fig. 12) in relation to neuronal and structural abnormality in the amygdala[43] and abnormality in amygdala connectivity[44], and resting-state cortical EEG[53]. Our findings are also consistent with past work indicating reduced prefrontal aromatase levels in individuals with ASD at postmortem[14,15]. The finding that BPA-associated gene methylation patterns in the BIS cohort were not sex-specific but that BPA-associated ASD symptoms and clinical diagnosis were more evident in males with a low genetic aromatase score would be consistent with the male vulnerability to BPA reflecting not differential epigenetic programming, but a greater vulnerability to reduced aromatase function in the developing male brain. This is reinforced by the ArKO model which resulted in an ASD-like phenotype in males not females. We have provided experimental evidence not only on the adverse neurodevelopment effects of BPA, but also experimental evidence of the alleviation of the behavioral, neurophysiological, and neuroanatomical defects following postnatal treatment with 10HDA. A human randomized controlled prevention trial that achieved bisphenol A elimination during pregnancy, with a resultant reduction in ASD among male offspring, would be a useful next step to provide further causal evidence of BPA risks but the feasibility and ethics of such an undertaking would be considerable. We demonstrate that postnatal administration of 10HDA may be a potential therapeutic agent that counteracts the detrimental impacts on distinct gene expression signatures directly impacted by prenatal BPA exposure. Furthermore, 10HDA may ameliorate deficits in ArKO mice which further suggests its utility as a replacement therapy for aromatase deficiency.

Two limitations of our human study were that BPA exposure was measured in only one maternal urine sample at 36 weeks, and that the assay may have low sensitivity[61]. We partially redressed the latter by focusing on categorical BPA values, as recommended[61], and undertook a matched ASD analysis where determinants for BPA variation, such as the urine collection time of day, were matched to reduce misclassification. Also, functional gene expression studies were unavailable for human samples in our study, but whilst the misclassification introduced by a reliance on a SNP based score would likely lead to an underestimation, an effect was still found among males with a low genetic aromatase score. It would be useful in further studies to consider altered aromatase function with a combined epigenetic-genetic score to reflect environment-by-epigenetic and genetic determinants of low aromatase function. Direct brain EWAS measures were not available, but for the key brain promoter PI.f region of CYP19A1, the brain-blood correlation is very high: Spearman's rho= 0.94, 95% CI [0.80, 0.98][32]. Although ASD symptoms (ASP score) at 2 years were based on parent report, we have previously reported that a higher ASP score was predictive of later ASD diagnosis by age 4[30]. ASD diagnosis at

analyses for the BPA-ASD cohort finding, and findings persisted after adjustment for further individual confounders. Using a modern causal inference technique, molecular mediation[60] we demonstrate in both birth cohorts that aromatase gene promoter I.f methylation underlies the known effect of higher prenatal BPA on BDNF hypermethylation. Other key features that support an underlying causal relationship include: the consistency of the findings across studies in this program

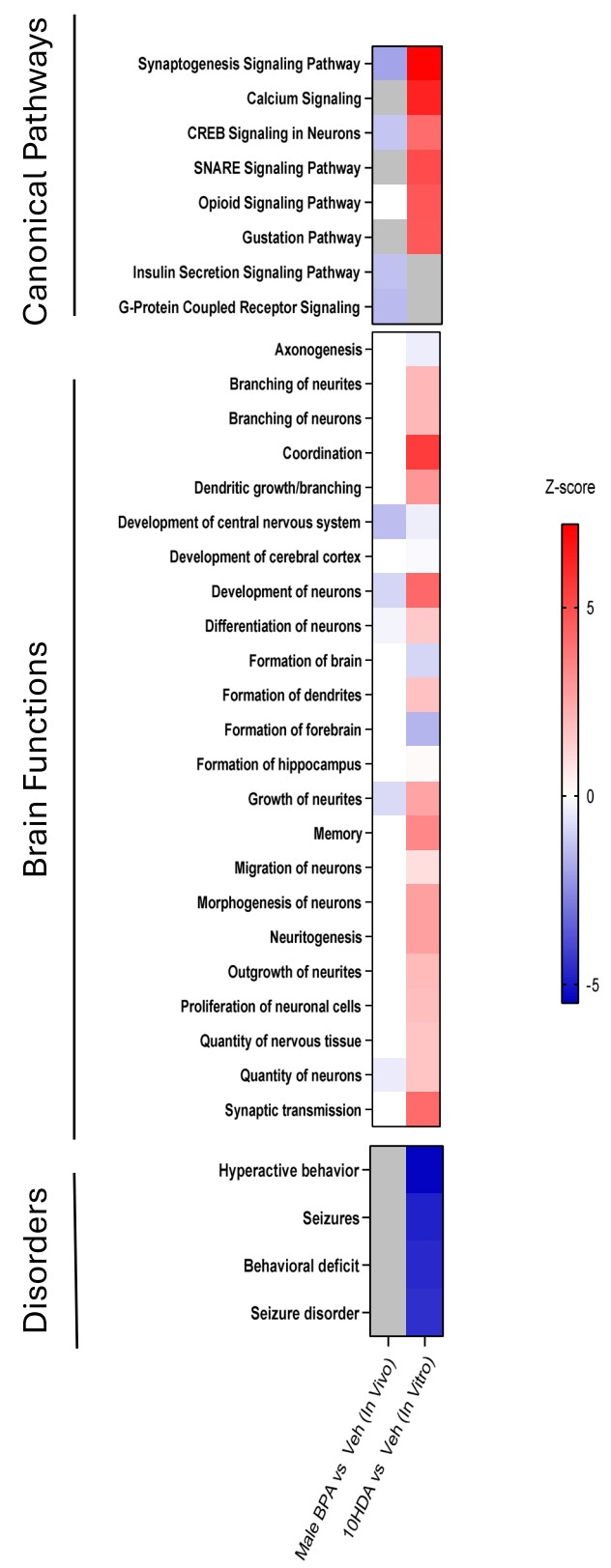

**Fig. 10 | Synaptogenesis pathway signaling, and growth of neurites are downregulated by BPA and upregulated by 10HDA.** The Canonical pathways and Disease and Function−Brain pathway databases were selected in Ingenuity. Several key signaling pathways and brain functions were downregulated (Z-score less than zero) by BPA and also upregulated (Z-score greater than zero) by 10HDA. 10HDA downregulated four brain disorder related pathways−hyperactive behavior, seizures, seizure disorder and behavioral deficit. Colored boxes indicate significant (P < 0.05, Fisher's Exact Test with Benjamani-hochberg) changes in z-score. Boxes shaded in gray indicate non-significant gene expression changes (P = 0.05 or greater). Source data are provided as a Source Data file.

on the cortex of fetal mice exposed to BPA in vivo, whereas 10HDA was performed in vitro on cortical primary cell culture. This would likely increase the variability between the BPA RNASeq and the 10HDA RNASeq, yet many of the same pathways were impacted but in opposing directions. Furthermore, the changes we observed in our RNAseq data could be due to changes in the cell type or cell state. This could be clarified by future single cell RNAseq experiments now that this specific issue has been identified.

The BPA exposure of mouse dams under our experimental conditions (50 μg/kg bodyweight) matches the current Oral Reference Dose set by the United States Environmental Protection Agencyc, the current safe level set by the U.S. Food and Drug Administration (FDA)[37], as well as the Tolerable Daily Intake set by the European Food Safety Authority[38] at the time that the mothers in our human cohort were pregnant[28]. The EFSA set a new temporary TDI of 4 μg/kg bodyweight in 2015[62] and, in December 2021, recommended further reducing this by five orders of magnitude to 0.04 ng/kg[63]; although this was subsequently revised to 0.2 ng/kg in EFSA's scientific opinion published in 2023[64]. Therefore, the timing of this new evidence is particularly pertinent and provides direct human data to support the reduced TDI.

Consistent with typical human exposure in other settings[20], BPA exposure in our birth cohort was substantially lower than the above, and yet we see adverse effects. Assuming fractional excretion of 1[65] and average daily urine output of 1.6 L[65], the median urinary bisphenol concentration of 0.68 μg/L−for which we see increased odds of ASD diagnosis−equates to a total daily intake of just 13 ng/kg, given a mean maternal bodyweight of 80.1 kg at time of urine collection. Notably, while we find an adverse association at 13 ng/kg, we do not have sufficient participants with lower exposure to evaluate a safe lower limit of exposure below this. Our findings in cell culture, with concentration 5 μg/L, parallels these human findings in terms of dose response. Although there are limitations in translating concentrations across body compartments without a stronger understanding of pharmacokinetics of BPA, 5 μg/L corresponds to the 90th and 95th percentile of BPA in urine in our human cohort, and allowing for a standard factor of 10 for variability in human sensitivity used when setting TDIs[38] indicates relevance down to at least 0.5 μg/L, below the median urine concentration. Our findings in laboratory animal studies, with exposure of 50 μg/kg bodyweight, are a little higher, as they were designed to correspond to the then current recommendations[36–38], but implications nevertheless have relevance within the range of exposure in our cohort. Allowing for standard factors of 10 for interspecies variability[38] and variability in human sensitivity[38], our animal study findings support a TDI at 500 ng/kg or below, which corresponds to the upper 0.5% of our human cohort. The findings of the human study, also allowing for a factor of 10 for variability in human sensitivity[38], therefore, support a TDI at or below 1.3 ng/kg.

Despite bans on its use in all infant products by the European Union in 2011 and the U.S. FDA in 2012, BPA remains widespread in the environment[66]. The main source of human exposure to BPA is dietary contamination[68]. Bisphenols are used in the production of common food contact materials, and migrate from those materials during use[69], including polycarbonate food and beverage containers and the epoxy

age 9 was verified to meet DSM-5 criteria by pediatrician audit of medical records, thereby reducing diagnostic misclassification.

In our preclinical studies, we performed the 10HDA studies and some of the BPA mechanistic studies only on male animals because our extensive laboratory and human studies, with more than 25 analyses, demonstrated that BPA exposure had significantly more adverse effects in males than females. In addition, we performed the RNAseq

linings of metal food cans, jar lids, and residential drinking water storage tanks and supply systems[64]. Additional sources of exposure include BPA-based dental composites and sealant epoxies, as well as thermal receipts[64]. BPA levels in pregnant women have previously been reported to be higher for young mothers, smokers, lower education, and lower income[70]. A substantial proportion of ASD cases might be prevented at the population level if these findings were causal and prenatal maternal BPA exposure were reduced. Here, exposures in the top quartile of BPA (>2.18 ug/L) correspond to a population attributable fraction (PAF) for males with low aromatase of 12.6% (95% CI 5.8%, 19.0%) although this estimate is imprecise as it is based on low case numbers. The only other available study with data on BPA exposure (>50 ug/L) and ASD provides an estimate in all children of 10.4%[67]. These studies have misclassification issues (e.g., a single urine measure for BPA and, in the Stein et al. study, an ASD diagnosis derived from health care sources[67]) but these misclassifications are likely non-differential and thus would bias findings towards the null. Additionally, we need to consider that the above findings of RfD/TDI and PAF are based on BPA alone. Factoring in that most exposures occur as part of a chemical mixture adds additional concern[71]. For example prenatal valproic acid exposed mice (an established ASD mouse model) also have a lower brain aromatase expression[72].

In summary, this multimodal program of work has shown an adverse effect of higher maternal prenatal BPA on the risk of male offspring ASD by a molecular pathway of reduced aromatase function, which plays a key role in sex-specific early brain development. Overall, these findings add to the growing evidence base of adverse neurodevelopmental effects from bisphenol and other manufactured chemical exposure during pregnancy. The case is compelling and supports broader evidence on the need to further reduce BPA exposure, especially in pregnancy. We also envision that our findings will contribute to new interventions for the prevention and/or amelioration of ASD targeting this specific pathophysiological pathway and we have identified one possible neuroprotective agent—10HDA—that has strong laboratory support. This agent now warrants further study, including human safety and efficacy evaluation.

## Methods
### Ethics
The human Barwon Infant Study cohort study was approved by the Barwon Health Human Research Ethics Committee, and families provided written informed consent. Parents or guardians provided written informed consent at prenatal recruitment and again when the child was 2 years of age. The human Columbia Center for Children's Environmental Health Mothers and Newborn cohort study was approved by the Institutional Review Boards of Columbia University and the Centers for Disease Control and Prevention, and all participants in the study provided informed consent. All procedures involving mice were approved by the Florey Institute of Neuroscience and Mental Health animal ethics committee and conformed to the *Australian National Health and Medical Research Council* code of practice for the care and use of animals for scientific purposes and All experiments were designed to minimize the number of animals used, as well as pain and discomfort. This work adheres to the ARRIVE essential 10 guidelines.

### Human studies
#### The Barwon Infant Study birth cohort
**Participants.** From June 2010 to June 2013, a birth cohort of 1074 mother–infant pairs (10 sets of twins) were recruited using an unselected antenatal sampling frame in the Barwon region of Victoria, Australia[28]. Eligibility criteria, population characteristics, and measurement details have been provided previously[28]; 847 children had prenatal bisphenol A measures available (Supplementary Table 1).

**Bisphenol A measurement.** We used a direct injection liquid chromatography tandem mass spectrometry (LC-MS/MS) method, as previously described in detail[73]. In summary, a 50 μL aliquot of urine was diluted in milli-Q water and combined with isotopically-labeled standards and b-glucuronidase (from *E. Coli*-K12). Samples were incubated for 90 min at 37 °C to allow for enzymatic hydrolysis of bisphenol conjugates before quenching the reaction with 0.5% formic acid. Samples were centrifuged before analysis, which was performed using a Sciex 6500 + QTRAP in negative electrospray ionization mode. The BPA distribution and quality control attributes for the application of this method to the Barwon Infant Study (BIS) cohort are shown in Supplementary Table 2.

**Child neurodevelopment.** Between the ages of seven and ten, a health screen phone call was conducted to gather information on autism spectrum disorder (ASD) diagnoses and symptomology. Out of the 868 individuals who responded to the health screen, 80 had an ASD diagnosis reported by their parents/guardians or were identified as potentially having ASD. The parent-reported diagnoses were confirmed by pediatric audit of the medical documentation to verify an ASD diagnosis as per DSM-5 guidelines. Participants that had a parent-reported diagnosis and then a verified pediatrician diagnosis by 30 June 2023 and whose diagnoses occurred before the date of their 9-year health screen were included as ASD cases in this study's analyses (n = 43). Participants were excluded if (i) their parent/guardian responded with 'Yes' or 'Under Investigation' to the question of an ASD diagnosis on the year-9 health screen but their diagnosis was not verified by 30 June 2023 (n = 26), or (ii) they had a verified diagnosis of ASD by 30 June 2023 but their date of diagnosis did not precede the date of their year-9 health screen (n = 15). The DSM-5-oriented autism spectrum problems (ASP) scale of the Child Behavior Checklist for Ages 1.5-5 (CBCL) administered at 2-3 years was also used as an indicator of autism spectrum disorder.

**Whole genome SNP arrays.** Blood from the umbilical cord was gathered at birth and then transferred into serum coagulation tubes (BD Vacutainer). Following this, the serum was separated using centrifugation as described elsewhere[74]. Genomic DNA was extracted from whole cord blood using the QIAamp DNA QIAcube HT Kit (QIAGEN, Hilden, Germany), following manufacturer's instructions. Genotypes were measured by Erasmus MC University Medical Center using the Infinium Global Screening Array-24 v1.0 BeadChip (Illumina, San Diego, CA, USA). The Sanger Imputation Service (Wellcome Sanger Institute, Hinxton, UK) was used for imputing SNPs not captured in the initial genotyping using the EAGLE2 + PBWT phasing and imputation pipeline with the Haplotype Reference Consortium reference panel[75]. Detailed methods are provided elsewhere[76].

Genome-wide DNA methylation arrays and analysis methods can be found in the Supplementary Methods.

**Center for Children's Environmental Health (CCCEH) epigenetic investigations.** The study participants consisted of mothers and their children who were part of the prospective cohort at the Columbia Center for Children's Environmental Health Mothers and Newborn (CCCEH-MN) in New York City (NYC). They were enrolled between the years 1998 and 2003, during which they were pregnant. The age range for these women was between 18 and 35, and they had no prior history of diabetes, hypertension, or HIV. Furthermore, they had not used tobacco or illicit drugs and had initiated prenatal care by the 20th week of their pregnancy. Every participant gave informed consent, and the research received approval from the Institutional Review Boards at Columbia University as well as the Centers for Disease Control and Prevention (CDC)[33].

Epigenetic methods have been previously described[77]. Briefly, DNA methylation was measured in 432 cord blood samples from the

CCCEH-MN cohort using the 450 K array (485,577 CpG sites) and in 264 MN cord blood samples using the EPIC array (866,895 CpG sites) (Illumina, Inc., San Diego, CA, USA).

BPA measures in the CCCEH were based on spot urine samples collected from the mother during pregnancy (range, 24–40 weeks of gestation; mean, 34.0 weeks)[33,78].

**Other statistical analysis.** Maternal urinary BPA concentrations were corrected for specific gravity to control for differences in urine dilution. Given a high proportion of the sample (46%) had BPA concentrations that were not detected or below the limit of detection (LOD), a dichotomous BPA exposure variable was formed using the 75th percentile as the cut-point. Dichotomizing the measurements in this way is also likely to give similar results regardless of whether indirect or direct analytical methods were used[79]. This is desirable since indirect methods might be flawed and underestimate human exposure to BPA[61].

To evaluate whether autism spectrum problems at 2 years could be used as a proxy for later ASD diagnosis, receiver operating characteristic (ROC) curve analyses were used. CBCL ASP at age 2 years predicted diagnosed autism strongly at age 4 and moderately at age 9 with an area under the curve of 0.92 (95% CI 0.82, 1.00) and 0.70 (95% CI 0.60, 0.80), respectively.

According to the normative data of the CBCL, T-scores greater than 50 are above the median. Due to a skewed distribution, ASP measurements were dichotomized using this cut point, which has respective positive and negative likelihood ratios of 2.68 and 0.00 in the prediction of verified ASD diagnosis at 4 years and 1.99 and 0.49 in the prediction of verified ASD diagnosis at 9 years.

A *CYP19A1* genetic score for aromatase enzyme activity was developed based on five genotypes of single nucleotide polymorphisms (CC of rs12148604, GG of rs4441215, CC of rs11632903, CC of rs752760, AA of rs2445768) that are associated with sex hormone levels[31]. Participants were classified as 'low activity' if they were in the top quartile, that is, they had three or more genotypes associated with lower levels of estrogen and as 'high activity' otherwise. Conditional logistic regression model analyses investigating the association between prenatal BPA levels and (i) early childhood ASP scores and (ii) verified ASD diagnosis at 9 years were conducted in the full sample, repeated after stratification by child's sex (assigned at birth based on visible external anatomy), and repeated again after further stratifying by the *CYP19A1* genetic score. Matching variables included child's sex (in the full sample analysis only), ancestry (all four grandparents are Caucasian vs not) and time of day of maternal urine collection (after 2 pm vs before). Within these matched groups, we additionally matched age-9 ASD cases and non-cases based on the date of the health screen and child's age at the health screen using the following procedure. Each case was matched to a single non-case based on nearest date of and age at health screen. Once all cases had one matched non-case, a second matched non-case was allocated to each case, and so on until all cases had 8 matched non-cases (8 was the most possible in the boys with high aromatase activity sub-sample and so this number was used across all sub-samples). The order by which cases were matched was randomly determined at the start of each cycle.

The guidelines for credible subgroup investigations were followed[80]. Only two categorical subgroup analyses were conducted, and these were informed a priori by previous literature and by initial mouse study findings. The adverse BPA effects in males with low aromatase enzyme activity (as inferred from the *CYP19A1* genetic score) were expected to be of higher magnitude, based on the prior probabilities from the laboratory work. A systematic approach was used to evaluate non-causal explanations and build evidence for causal inference, considering pertinent issues such as laboratory artefacts that are common in biomarker and molecular studies[81].

A second CYP19A1 genetic score was developed for use in sensitivity analyses. The Genotype-Tissue Expression (GTEx) portal was used to identify the top five expression quantitative trait loci (eQTLs) for aromatase in any tissue type that showed a consistent effect direction in brain tissue. A functional genetic score was then computed for each BIS participant by summing the number of aromatase-promoting alleles they carry across the five eQTLs (AA of rs7169770, CC of rs1065778, AA of rs28757202, CC of rs12917091, AA of rs3784307), weighted by their normalized effect size (NES) in amygdala tissue. The score was then reversed so that higher values indicate lower aromatase activity. The score thus captures genetic contribution to reduced cross-tissue aromatase activity with a weighting towards the amygdala, a focus in our animal studies. The variable was dichotomized using the 75th percentile as the cut-point and the above stratified analyses were repeated with this new weighted score replacing the original, unweighted score.

For the human epigenetic investigations, we used multiple linear regression and mediation[60] approaches. As in past work[33], BPA was classified as greater than 4 μg/L vs less than 1 μg/L in the CCCEH-MN cohort. A comparable classification was used for the BIS cohort, with greater than 4 μg/L vs the rest. In both cohorts the regression and mediation analyses were also adjusted for sex, gestational age, self-reported ethnicity, and cord blood cell proportions. In the BIS cohort, ethnicity was defined as all four grandparents are Caucasian vs not (see Table S3). For the CCCEH-MN cohort, ethnicity was defined as Dominican vs African American[33]. We used statistical software packages R v3.6.3[82] and Stata 15.1[83].

## LABORATORY STUDIES
### SHSY-5Y cell culture study
**BPA treatment on aromatase expression in cell culture.** Human neuroblastoma SHSY-5Y cells were chosen because they were known to express aromatase and SH-SY5Y have been used in ASD research[84]. SHSY-5Y cells (CRL-2266, American Type Culture Collection, Virginia, USA) were maintained in Dulbecco's Modified Eagle Medium (DMEM) (10313-021, Gibco-life technologies, New York (NY), USA) supplemented with 10% heat-inactivated Fetal Bovine Serum (FBS) (12003C-500 mL, SAFC Biosciences, Kansas, USA), 1% penicillin streptomycin (pen/strep) (15140-122, Gibco-life technologies, NY, USA) and 1% L-Glutamine (Q) (25030-081, Gibco-life technologies, NY, USA) at 37 °C in a humidified atmosphere of 95% air and 5% $CO_2$. SHSY-5Y cells were grown in 175 cm$^2$ cell culture flasks (T-175) (353112, BD Falcon, Pennsylvania, USA). Cells were passaged when the seeding density of the T-175 flasks was reached (roughly 80-90% confluence). Cells were passaged by aspirating media from flasks and flasks were then washed once with 10 mL of DPBS (14190-136 Gibo-life technologies, NY, USA) to remove the FBS (inhibits the actions of trypsin). Next, cells were incubated with trypsin (2 mL/T-175 flask) at 37 °C for 5 min to detach cells from the flask wall. To prevent further action of trypsin, media (8 mL/T-175 flask) was added, and contents were pipetted up and down to disperse cell clumps. The cell suspension was then transferred to a 15 mL centrifuge tube (430791, Corning CentriStar, Massachusetts, USA) and centrifuge (CT15RT, Techcomp, Shanghai, China) for 5 min at 1000 RPM at room temperature (RT). The media was then aspirated from these tubes and the cell pellet resuspended in 1 mL of media. Cell viability counts were performed using a hemocytometer (Hausser Scientific, Pennsylvania, USA) to determine the number of live versus dead cells in solution. Two μL of cell suspension was diluted with media (98 μL) and then trypan blue (100 μL) (T8154, Sigma-Aldrich Co., St. Louis, MO, USA) (which labeled dead cells) in a sterile microcentrifuge tube (MCT-175-C-S, Axygen, California, USA). Ten μL of this solution was loaded into the hemocytometer and imaged using a light microscope (DMIL LED, Leica, Germany). Dead cells appeared blue under the microscope because these cells take up the dye whereas live cells were clear (i.e., unstained). Cells were counted in the outer four

squares located in each chamber (two chambers, eight squares), with their dimensions known. The average of the eight counts was multiplied by the dilution factor and by 104, yielding the concentration of cells/mL solution. Average cell counts were plotted against treatment groups using GraphPad Prism 9.0 (GraphPad Software, Inc., San Diego, CA).

**Chemicals.** Bisphenol A (BPA) (239658-50 G, Sigma-Aldrich Co., St. Louis, MO, USA) was used for cell treatment. Prior to treatment, stock solutions of each drug were prepared as stated below. BPA was dissolved in pure ethanol (EA043-2.5 L, Chem supply, South Australia, Australia) and the final concentration of the stock solutions was 0.0435 g/mL. Cells in T-175 flasks were randomly assigned to receive treatment with BPA at a dosage of 100 µg/L, 50 µg/L, 25 µg/L or 0 µg/L (vehicle). There was also a no treatment (no vehicle added) flask.

Cell treatment, protein assay, SDS-Page, and western blotting methods can be found in the Supplementary Methods.

## Animal studies
**Animals.** Two colonies of mice, maintained at the Florey Institute, were used in this study. The Aromatase knockout (ArKO) mouse model and the Aromatase-enhanced fluorescent green protein (Cyp19-EGFP) transgenic mouse model. Animals were monitored daily except for weekends. If animals showed general clinical signs, an animal technician or a vet was consulted for advice and euthanasia performed as required.

Mice were maintained under specific pathogen-free (SPF) conditions on a 12 h day/night cycle, with *ad libitum* water and soybean-free food (catalog number SF06-053, Glen Forrest Stockfeeders, Glen Forrest, Western Australia, Australia). Facial tissues were provided for nesting material, and no other environmental enrichment was provided. The room temperature ranged from 18 °C-23 °C and the humidity ranged from 45%-55%.

The sex of mice was determined by SRY genotyping if fetal, otherwise sex was determined by examining the anogenital region around PND9 and again at weaning. Sex was confirmed by inspecting the gonads during dissection. *Cyp19*-EGFP mice were toe and tail clipped for identification and genotyping at PND9, ArKO mice were ear notched and tailed clipped at two weeks of age. The oligonucleotide sequences (custom oligos, Geneworks, Australia) for ArKO, GFP and SRY genotyping can be found in Supplementary Data File 2.

**Aromatase knockout (ArKO) mouse model.** The ArKO mouse is a transgenic model having a disruption of the *Cyp19a1* gene. Exon IX of the *Cyp19a1* gene was replaced with a neomycin-resistant cassette[41]. Homozygous Knockout (KO) and wild-type (WT) offspring were bred by mating heterozygous (het) ArKO parents and then PCR genotyped. ArKO mice were backcrossed onto a C57BL/6 J background strain, >10 generations (obtained from Animal Resources Centre, Western Australia) and the colony maintained at the Florey institute.

**Aromatase-enhanced fluorescent green protein (*Cyp19*-EGFP) transgenic mouse model.** The Cyp19-EGFP mouse model (backcrossed onto the FVBN background strain >10 generations, obtained from Animal Resources Centre, Western Australia) is a transgenic model having a bacterial artificial chromosome containing the full length of the *Cyp19a1* gene with an Enhanced Green Fluorescent Protein (EGFP) gene inserted upstream of the ATG start codon[11]. Thus, EGFP expression is an endogenous marker for *Cyp19a1* expression. This allows for the visualization and subsequent localization of EGFP as the marker for aromatase without the use of potentially nonspecific aromatase antibodies[11]. We have previously characterized this transgenic model and its brain expression of EGFP[11]. Based on our characterization studies, this transgenic model does not have phenotypes that are significantly different to wildtype mice.

**Early postnatal 17β-estrodiol treatment.** Mice were allocated into three groups: (1) WT mice receiving a sham implantation; (2) ArKO mice receiving a sham implantation; and (3) ArKO mice undergoing implantation with a 17β-estradiol pellet (sourced from Innovative Research America). This estradiol pellet was designed to release 0.2 mg of 17β-estradiol steadily over a period of 6 weeks. A corresponding sham pellet, identical in size but devoid of E2, was implanted in the control groups.

The implantation procedure was carried out on postnatal day 5. For anesthesia, mice were exposed to 2% isoflurane (IsoFlo, Abbott Laboratories, VIC, Australia) within an induction chamber. The efficacy of anesthesia was confirmed by the lack of response to foot-pinch stimuli. During the surgical procedure, mice were maintained on a heated pad to regulate body temperature. A small, 5 mm incision was made in the dorsal region for the subcutaneous insertion of the pellet, preceded by an injection of Bupivacane in the same area. Following the implantation, the incision was carefully sutured. Post-surgery, mice were placed in a thermal cage (Therma-cage, Manchester, UK) for recovery and monitoring until they regained consciousness and could be returned to their respective litters. Any mice exhibiting complications such as opened stitches were excluded from the study.

**BPA injection administration treatment.** Plugged FVBN dams were randomly assigned, blocking by weight gain at E9.5 and litter/cage where applicable, to receive daily scruff subcutaneous injections (24 G x 1", Terumo, Somerset, New Jersey, USA) of BPA (239658-50 G, Sigma-Aldrich Co., St Louis, MO, USA) in ethanol and peanut oil (Coles, Victoria, Australia), either between E0.5-E9.5, E10.5-E14.5 or E15.5-birth at a dosage of 50 µg/kg (deemed as the safe consumption level by the Food and Drug Administration, FDA)[37] or 0 µg/kg (vehicle) of maternal body weight. The injection volume was 1.68 µL/g bodyweight. Mice were weighed directly before each injection. BPA and vehicle exposed litters did not differ in litter size (Supplementary Fig. 14).

**10HDA injection administration treatment.** *Cyp19*-EGFP or ArKO mice were randomly assigned by blocking on sex and litter to receive daily intraperitoneal injections (31 G x 1", Terumo, Somerset, New Jersey, USA) of 500 µg/kg 10HDA (Matreya, USA) in saline or vehicle saline for 21 consecutive days. The injection volume was 2.1 µL/g bodyweight. Mice were weighed directly before each injection.

**BPA oral administration treatment.** Plugged FVBN dams were exposed to jelly at E9.5. The jelly contained 7.5% Cottee's Raspberry Cordial (Coles, Victoria, Australia) and 1% bacteriological agar (Oxoid, Australia) in milli-Q water. The pH was increased to between 6.5-7.5 with a pallet of NaOH to allow the jelly to set. Dams were then randomly assigned, blocking by weight gain at E9.5 and litter/cage where applicable, to receive a daily dose of jelly, which contained either ethanol or BPA dissolved in ethanol, at a dosage of 50 µg/kg or 0 µg/kg (vehicle) of maternal body weight. Dams received doses between E10.5-E14.5, and only dams that were observed to have consumed all the jelly each day were included in the study.

## Behavioral paradigms
**Three-chamber social interaction test.** The three-chamber social interaction test is extensively used to investigate juvenile and adult social interaction deficits, including in sociability[85,86]. BPA exposed pups were habituated in the experimental room on P21, directly after weaning. Following a two-to-three-day habituation, testing was conducted from P24 to P27, as only a maximum of ten mice could be tested during the light phase per day. ArKO mice treated with estrogen or sham pallet began habituation at PND28-29, with testing at PND31-33. Both male and female mice were tested. Testing was performed in a dedicated room for mouse behavior studies; no other animals were

present in the room at the time of acclimatization and testing. The temperature of the room was maintained at approximately 21 °C.

The test apparatus, a three-chambered clear plexiglass, measuring 42 cm x 39 cm x 11 cm, had two partitions creating a left, right (blue zones), and center chamber (green zone) in which mice could freely roam via two 4 cm x 5 cm openings in the partitions (Supplementary Fig. 15). The two side chambers contained two empty wire cages. A 1 cm wide zone in front of each wire cage was defined as the interaction zone (yellow zones). The chamber was set on a black table for white mice, and on a white covering for black mice to aide tracking.

Each test consisted of two consecutive 10-min trials, a habituation trial (T1) and a sociability trial (T2). T1 allowed the test mouse to habituate, and any bias for either empty interaction zone was noted. For T2, a C57BL/6 J novel stranger mouse matched with the test mouse for age and gender was introduced into the cage on the opposite side to which the test mouse demonstrated an interaction zone bias. Thus, any evidence of sociability is bolstered as interaction zone bias would have to be overcome.

For each trial, the test mouse began in the center chamber, and its activity, both body center point, and nose point was tracked and quantified by TopScan Lite (Clever Sys Inc., Reston VA, USA). In this study, the key measure extracted was the average duration of the nose point in each interaction zone.

Social approach and sociability were analyzed. We define social approach as the time the test mouse's nose point was tracked in the stranger cage interaction zone. Sociability is the higher proportion of time the test mouse to spends with the nose point in the stranger cage interaction zone compared to the empty cage interaction zone.

Details on the Y-maze and grooming methods can be found in the Supplementary Methods.

**Golgi staining.** Mice had not undergone any behavioral testing. For Golgi staining and analysis, Wild Type (WT) and Knockout (ArKO) and *Cyp19*-EGFP littermate males (aged P65-P70); one mouse from $n = 3$ litters for each genotype) were deeply anesthetized with isopentane rapidly decapitated and fresh whole brain tissues were collected. Brains were first washed with milli-Q water to remove excess blood and then directly placed in the solution obtained from the FD Rapid GolgiStain™ Kit (FD Neuro-Technologies, Inc., MD, USA). Brains were stored at room temperature in the dark and the solutions were replaced after 24 hours, and the tissues were kept in the solution for two weeks. After two weeks, tissues were transferred into solution C for a minimum of 48 hours at room temperature. For sectioning, brains were frozen rapidly by dipping into isopentane pre-cooled with dry ice, and 100 μm thick coronal sections were cut at -22 °C and mounted on 1% gelatin-coated slides. The sections were then air dried in the dark at room temperature. When sections were completely dry, slides were further processed and rinsed with distilled water and placed in the solution provided in the kit for 10 min and washed again with distilled water followed by dehydration for 5 min each in 50%, 75%, 95%, and 100 % ethanol. Sections were further processed in xylene and mounted with Permount.

**Neuron Tracing.** Neuron tracing was conducted on the amygdala and somatosensory cortex of BPA-exposed mice (exposed ED10.5-14.4) and untreated ArKO mice. Neuron tracing in the amygdala was conducted in both male and female mice, and in the somatosensory cortex, only in male mice. Stained slides were coded to ensure that morphological analysis was conducted by an observer who was blind to the animals' treatment. Morphological analysis followed a previously described protocol[87] with the following modifications: layer V pyramidal cells of the somatosensory cortex, which were fully impregnated and free of neighboring cells or cellular debris, were randomly selected for analysis (Supplementary Fig. 16). Golgi-stained coronal sections containing medial amygdala and somatosensory

cortical area were visualized under Olympus BX51 microscope. Neuronal tracing was carried out with the help of Neurolucida and Neuroexplorer software (MicroBright Field Inc., Williston, USA). Up to three pyramidal cells in the MeA and four pyramidal cells in the somatosensory cortex per section over 3 sections (9 (MeA) and 12 (cortex) cells per animal respectively) were sampled[88,89]. For Sholl analysis[90], concentric circles were placed at 10 μm intervals starting from the center of the cell body and the parameters i) total dendritic length (sums of the length of individual branches) of apical and basal dendrites of pyramidal cells and ii) number of spines (protrusions in direct contact with the primary dendrite) and their density (number of spines per 10 μm) were recorded.

*Neuron selection criteria:* Neurons were selected based on the following criteria. They had to be fully stained, and the cell body had to be in the middle third of the section thickness. The dendrites of the neuron had to be unobscured by the other nearby neuron. Also, neurons had to possess tapering of the majority of the dendrites towards their ends. Representative images of neurons from vehicle and BPA-exposed adult mice can be found in Supplementary Fig. 17.

**Visualizing c-Fos activation to conspecific exposure (amygdala) Stranger exposure paradigm procedure.** Cyp19-EGFP mice of both sexes as well as male ArKO mice, together with male WT littermates were utilized in this study. Mice had not undergone any other behavior testing. All test mice were acclimatized to the testing room in individual cages for three nights prior to testing. All mice were age P24 on the day of testing, which was performed between 10 am-2 pm. Testing was performed in a dedicated room for mouse behavior studies and no other animals were present in the room at the time of acclimatization and testing. The temperature of the room was maintained at approximately 21 °C.

On the day of testing, each mouse cage containing the isolated test mouse was placed on a stage (a trolley). The lid containing food and water was removed and immediately following, a sex-/age-matched C57Bl/6 J stranger mouse or a novel object (new 1 mL syringe) was placed into the cage and a clean, empty lid was placed on the top. New gloves were used to handle each syringe to avoid transferring another mouse's olfactory signature to it. The 10 min trial began as soon as the cage lid was shut. After 10 min had elapsed, the stranger or the novel object was removed, the test mouse with home cage was returned to its original location with the original cage lid with food and water for 2 hours prior to perfusion. Once it was established that there was a difference in c-fos expression between stranger exposure and novel object exposure in the medial amygdala, BPA and vehicle exposed (ED10.5-14.5) Cyp19-EGFP mice as well as estrogen and sham pallet treated ArKO and WT mice were exposed to an age and sex matched stranger as described above. C-fos expression was quantified in male mice only.

Histology and stereological analysis methods can be found in the Supplementary Methods.

Neuron count brain collection, staining, brain region delineation and stereology can be found in the Supplementary Methods.

**Electrophysiological studies**
**Microelectrode array electrophysiology.** Male mice aged 8 weeks weighing between 15 and 20 g were used for this study. They had not undergone any behavioral testing prior to electrophysiology. We studied synaptic activity parameters such as the Input/Output (I/O) curve. Stimulation of the glutamatergic synapses terminate in the basolateral amygdala (BLA) and the basomedial amygdala (BMA), which were integrated with multiple inputs that compute to produce an output (field excitatory postsynaptic potential, fEPSP). I/O curve serves as an index of synaptic excitability of large neuronal populations. Mice were anesthetized with isoflurane (IsoFlo™, Abbott Laboratories, Victoria, Australia) and decapitated. The whole brains were quickly removed

and placed in ice-cold, oxygenated (95% $O_2$, 5% $CO_2$) cutting solution (composition in mmol/L: 206 sucrose, 3 KCl, 0.5 $CaCl_2$, 6 $MgCl_2$-$H_2O$, 1.25 $NaH_2PO_4$, 25 $NaHCO_3$, and 10.6 D-glucose). Coronal brain amygdala slices (300 μm) were prepared with a VT 1200 S tissue slicer (Leica) and quickly transferred to 34 °C carbogen bubbled artificial CSF (aCSF) (composition in mmol/L: 126 NaCl, 2.5 KCl, 2.4 $CaCl_2$, 1.36 $MgCl_2$-$H_2O$, 1.25 $NaH_2PO_4$, 25 $NaHCO_3$, and 10 D-glucose) for 30 min. After further recovery of 1 h equilibrium in oxygenated aCSF at room temperature, the slices were transferred to a submission recording chamber, an MEA chip with 60 electrodes spaced 200 μm apart (60 MEA 200/30 iR-Ti: MCS GnbH, Reutlingen, Germany). The slice was immobilized with a harp grid (ALA Scientific Instruments, New York, USA) and was continuously perfused with carbogenated aCSF (3 mL/min at 32 °C). fEPSPs produced in BLA and BMA were by stimulation of a randomly chosen electrode surrounding the target area with a biphasic voltage waveform (100 μs) at intermediated voltage intensity. The electrode could only be chosen if it produced a fair number of fEPSPs in the surrounding recording electrodes. The width of the EPSP wave ranged from 20 to 30 ms was selected. We chose slices where BLA and BMA were greatly represented according to Allen Mouse Brain Atlas[91]. Care was taken to choose the stimulating electrode in the same region from one slice to the other. The peak-to-peak amplitude of fEPSP in BLA and BMA was recorded by a program of LTP-Director and analyzed using LTP-Analyzer (MCS GnbH, Reutlingen, Germany).

### Electrocorticogram (ECoG)
**Electrocorticogram recordings.** Male mice aged 8 weeks were used for this study. Mice had not undergone any behavioral testing prior to ECoG recording. For ECoG, surgeries were performed as previously described[92]. Mice were anesthetized with 1–3% isoflurane and two epidural silver 'ball' electrodes implanted on each hemisphere of the skull. Electrodes were placed 3 mm lateral of the midline and 0.5 mm, caudal from bregma. A ground electrode was placed 2.5 mm rostral from bregma and 0.5 mm lateral from the midline. Mice were allowed to recover for at least 48 hours after surgery. ECoGs were continuously recorded in freely moving mice for a 4–6-hour period during daylight hours following a standard 30-min habituation period. Signals were band-pass filtered at 0.1 to 40 Hz and sampled at 1 kHz using the Pinnacle EEG/EMG tethered recording system (Pinnacle Technology Inc, KS). Power spectrums were calculated using Hann window with a resolution of 1 Hz using Sirenia Pro analysis software (Pinnacle Technology Inc) on stable 30-min periods of ECoG recordings.

### Primary Cortical Cultures
**Neuroprotective effect of 10HDA against injury induced by BPA on embryonic mouse cortical neurons.** Primary cortical neurons were obtained from male *Cyp19*-EGFP mouse embryos at gestational day 15.5. Embryos were genotyped for SRY to determine sex, and only male embryos were used. Cells were seeded in 24-well plates containing 12 mm glass coverslips, coated with 100 μg/mL poly D-lysine to a density of ~0.45 x $10^6$ cells/well and incubated in a humidified $CO_2$ incubator (5% $CO_2$, 37 °C). Cells were pre-treated with vehicle (DMSO), 1 mM 10HDA (Matreya, PA,USA), 25 nM BPA and 1 mM 10HDA with 25 nM BPA. For each group, 10 neurons were measured, and the experiments were duplicated. Each replicate was from a separate culture.

Cells were fixed in 4% paraformaldehyde and stained with mouse anti-βIII tubulin monoclonal primary antibody (1:1000; cat #ab41489, Abcam, United Kingdom) and goat anti-mouse secondary antibody, Alexa Fluor 488, (1:2000; cat#A11017; Invitrogen, USA) to label neuronal cells. Aromatase was stained using Rabbit anti-aromatse Antibody (1:2000 cat# A7981; Sigma Aldrich, St. Louis, MO, USA) and donkey anti-rabbit Alexa594 (1:2000; cat# A-21207; Invitrogen, USA). Cell nucleus was stained with Hoechst 33258 solution (Sigma 94403

(2 μg/mL)). Images were captured using an Olympus IX51 microscope (X40 objective). Neurites were quantified using Neurolucida and Neuroexplorer software (MicroBright Field Inc, Williston, USA) as described in the neuron tracing section.

### RNAseq
**RNA extraction.** Total RNA was extracted using PARIS kit (cat#: AM1921, Invitrogen™PARIS™ Kit) according to the protocol supplied by the manufacturer. cDNA libraries were generated using the SureSelect.

Strand-Specific RNA Library Prep for Illumina Multiplexed Sequencing kit (Agilent Technologies, CA, USA), according to manufacturer's instructions.

**In vivo effects of BPA on Fetal brain cortical RNA seq.** Pregnant Cyp19-EGFP dams were injected subcutaneously with BPA or vehicle ED10.5-14.5 as described in previous section, and culled on ED15.5 by isoflurane overdosed. Fetuses were harvested and placed in chilled PBS. Embryo brain cortical tissue was dissected from fetuses, snap frozen in liquid nitrogen and stored in −80°C until RNA extraction. The sex of fetuses was determined by visual assessment of the gonads and *Sry* (a male-specific gene) genotyping. Each RNA seq run, 6 cDNA libraries (derived from total RNA samples with 3 biological samples per group), were analyzed by MidSeq Nano run, 50 bp, Single end read on the Illumina platform. Because of undetectable levels of *Cyp19a1* RNA in the fetal brain, *Cyp19a1* RNA levels were not included. This is consistent in that Aromatase+ cells represent <0.05% of neurons in the adult mouse brain[93]. Subsequent in vivo transcriptomic analyses were completed in males only. Read quality was then assessed with FastQC. The sequence reads were then aligned against the Mus musculus genome (Build version GRCm38). The Tophat aligner (v2.0.14) was used to map reads to the genomic sequences. Sequencing data were then summarized into reads per transcript using Feature counts[94]. The transcripts were assembled with the StringTie tool v1.2.4 using the reads alignment with Mus_musculus.GRCm38 and reference annotation based assembly option (RABT) using the Gencode gene models for the mouse GRCm38/mm10 genome build. Normalisation and statistical analysis on the count data were executed using EdgeR (version edgeR_3.14.2 in R studio, R version 3.14.2). The data were scaled using trimmed mean of M-values (TMM)[95] and differentially expressed genes between all treatment group (Benjamini−Hochberg false discovery rate >0.1). Differentially expressed genes (DEGs) were identified by comparing mice exposed to 50 μg/kg/day BPA with those exposed to the vehicle.

**In vitro effects of 10HDA in primary cell culture RNASeq.** Primary brain cortical neurons were obtained from C57BL/6 mouse embryos at GD 15.5. Neuronal cell cultures were treated with vehicle (DMSO) or 1 mM 10HDA (Matreya, PA,USA) as described above. The libraries were sequenced with 50 bp single end reads using an Illumina Hiseq and read quality assessed using FastQC. Untrimmed reads were aligned to mouse mm10 genome using Subjunc aligner (version 1.4.4) within the Subread package[96]. Sequencing data were then summarized into reads per transcript using Feature counts[94] and the Gencode gene models for the mouse GRCm38/mm10 genome build (August 2014 freeze)[97]. Normalisation and statistical analysis on the count data were executed using EdgeR (version edgeR_3.4.2 in R studio, R version 3.0.2)[98] after removing features with less than 10 counts per million for at least 3 of the samples. The data were scaled using trimmed mean of M-values (TMM)[95] and differentially expressed genes between all treatment group (Benjamini−Hochberg false discovery rate >0.1). Annotation was added using the ensemble mouse gene annotation added using bioMart package[99]. Differentially expressed genes were identified by comparing cells exposed to 10HDA with those exposed to the vehicle.

**Pathway analysis.** The BPA and 10HDA differential expression data for enriched pathways were analyzed using Ingenuity (QIAGEN) and tested against the Canonical Pathway Library, Brain Diseases and Functions Library and the Brain Disorders pathway libraries. We included the top 8 Canonical pathways. Then we included only pathways which were $p < 0.05$ in both the BPA and the 10HDA data for the Brain Diseases and Functions pathway libraries, and included all $P$-values for the Brain Disorders pathway library ($p > 0.05$ are in gray).

An additional analysis of the gene expression data was performed using the *clusterProfileR* R package[100], which provides a range of statistical tests to detect pathways from a query gene set. The test used here was Gene Set Enrichment Analysis (GSEA)[101], and the genes were tested against the Gene Ontology pathway database (specifically, GO: Biological Process)[102].

**Computational Molecular Docking.** The DockThor molecular docking platform[103] was used to assess binding affinities between estrogen receptor beta (encoded by *ESR2* gene) and the ligands 17-beta estradiol (E2; the native ligand), BPA, and 10HDA. DockThor takes as input 3D molecular structures for a putative receptor-ligand pair and employs a genetic-algorithm-based optimization strategy to identify optimal binding position within a specified search region. The crystal structures of estrogen receptor alpha (Erα) and beta (Erβ) were sourced from the Protein Data Bank (PDB) with respective PDB IDs: Erα - 5KRI and Erβ - 1YYE. For each ligand, the search grid was restricted to the known estrogen receptor beta ligand-binding domain, centered at $x = 30$, $y = 35$, $z = 40$, with total grid size of $x = 25$, $y = 28$ and $z = 22$. Default settings were used for the optimization procedure.

**Statistical analysis.** Researchers were blind to treatment during the conduct of the experiment and the outcome assessment but not during statistical analysis.

Mean, standard deviation (SD), and standard error of the mean (SEM) were calculated with GraphPad Prism version 9.4 (GraphPad Software).

Data were tested for equal variances and normality using the Shapiro Wilk test. As electrophysiology, Golgi staining and primary cell culture experiments utilized several data points per animal, observations were not independent, and this non-independence was accounted for in our analyses. We used generalized estimating equations (GEEs) in R version 4.1.2. in a marginal modeling approach that estimates population-averaged effects while treating the covariance structure as a nuisance. We specified the covariance structure as exchangeable (that is, assumed equal correlation between pairs of measurements on the same animal). Given the small number of clusters (i.e. animals), bootstrapped standard errors were estimated using 200 repeats to maintain a conservative type 1 error rate[104]. An interaction term was added to the amygdala Golgi staining study to assess a sex * genotype or sex * BPA exposure interaction.

Where data were normally distributed, parametric tests were conducted. For more than two groups, a one-way ANOVA was conducted with Holm-Sidak post hoc FDR correction, with alpha set to 0.05. Otherwise, unpaired two-tailed Student's $t$-tests were used to compare two variables. In cases where normality was not assumed, a Mann-Whitney (comparing two groups) or Kruskal-Wallis with Dunn's post hoc (comparing three or more groups) was used. In the case of the three-chamber data, a two-way mixed ANOVA was used to assess group x cage side interaction (stranger cage interaction zone vs empty cage interaction zone) with post hoc testing adjusted by the Holm-Sidak method.

Comparisons made are indicated on the Figure legends, and $p$-values $< 0.05$ were considered significant. All tests were two-sided (two-tailed) where applicable.

## Reporting summary

Further information on research design is available in the Nature Portfolio Reporting Summary linked to this article.

## Data availability

The BIS data including all data used in this paper are available under restricted access for participant privacy. Access can be obtained by request through the BIS Steering Committee by contacting Anne-Louise Ponsonby, The Florey institute of Neuroscience and Mental Health, annelouise.ponsonby@florey.edu.au. Requests to access cohort data will be responded to within two weeks. Requests are then considered on scientific and ethical grounds and, if approved, provided under collaborative research agreements. Deidentified cohort data can be provided in Stata or CSV format. Additional project information, including cohort data description and access procedure, is available at the project's website https://www.barwoninfantstudy. org.au. Source data underlying Figs. 1–6, 8–10 and Supplementary Figs. 2, 3–7,14 have been provided as a Source Data file with this paper. The RNAseq data discussed in this publication have been deposited in NCBI's Gene Expression Omnibus[105] and are accessible through GEO Series accession numbers; fetal brain expression with and without prenatal BPA exposure, GSE266401 and primary cortical culture treated with and without 10HDA, GSE266400 Source data are provided with this paper.

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

## Acknowledgements

The authors thank the BIS participants for their generous contribution to this project. The authors also thank current and past cohort staff. The establishment work and infrastructure for the BIS was provided by the Murdoch Children's Research Institute, Deakin University, and Barwon Health, supported by the Victorian Government's Operational Infrastructure Program. We thank all the children and families participating in the study, and the BIS fieldwork team. We acknowledge Barwon Health, Murdoch Children's Research Institute, and Deakin University for their support in the development of this research. We thank Dr Shanie Landen for statistical advice, and Alex Eisner for independent statistical review of the analyses in the manuscript. We thank Soumini Vijayay and Kristie Thompson for human BPA lab measurement and Dr Steve Cheung for assistance preparing the primary cortical culture. We thank Chitra Chandran, Georgia Cotter, Stephanie Glynn, Oliver Wood and Janxian Ng for manuscript preparation. Manuscript editor Julian Heng (Remotely Consulting, Australia) provided professional editing of this article. This multimodal project was supported by funding from the Minderoo Foundation. Funding was also provided by the National Health and Medical Research Council of Australia (NHMRC), the NHMRC-EU partnership grant for the ENDpoiNT consortium, the Australian Research Council, the Jack Brockhoff Foundation, the Shane O'Brien Memorial Asthma Foundation, the Our Women's Our Children's Fund Raising Committee Barwon Health, The Shepherd Foundation, the Rotary Club of Geelong, the Ilhan Food Allergy Foundation, GMHBA Limited, Vanguard Investments Australia Ltd, and the Percy Baxter Charitable Trust, Perpetual Trustees, Fred P Archer Fellowship; the Scobie Trust; Philip Bushell Foundation; Pierce Armstrong Foundation; The Canadian Institutes of Health Research; BioAutism, William and Vera Ellen Houston Memorial Trust Fund, Homer Hack Research Small Grants Scheme and the Medical Research Commercialisation Fund. This work was also supported by Ms. Loh Kia Hui. This project received funding from a NHMRC-EU partner grant with the European Union's Horizon 2020 Research and Innovation Programme, under Grant Agreement number: 825759 (ENDpoiNTs project). This work was also supported by NHMRC Investigator Fellowships (GTN1175744 to D.B., APP1197234 to A.-L.P., and GRT1193840 to P.S.). The study sponsors were not involved in the collection, analysis, and interpretation of data; writing of the report; or the decision to submit the report for publication.

## Author contributions

Conceptualization—laboratory experiments: W.C.B., N.R., B.T., L.M.D.D. Laboratory experiments and analysis: W.C.B., K.V., S.D., H.K.C., J.C., F.C., CR, T.K., G.B.B., A.E.-B., S.E.H., N.T.T., K.B. Supervision of lab data collection: W.C.B., K.V., S.D., C.R. Laboratory statistical analysis: W.C.B., K.V., F.C., C.R., S.Th., V.H., Y.W.Y. Design and conduct of the Barwon Infant Study: C.S., A.-L.P., P.V., D.B., P.S., C.L., M.L.K.T., BIS Investigator Group. Design, conduct and analysis of the CCCEH-MN study: J.B.H., S.W., J.G. Design and conduct of BPA study measures in BIS: J.M., C.S., A.-.L.P. Design, conduct, and analysis of gene methylation studies: S.Ta., B.N., T.M., R.S., D.D., A.-L.P. Human studies statistical analysis: C.S., S.Th., A.-.L.P., S.Ta., K.V., M.O.H. Writing—reports and original draft: C.S., K.V., S.Th., S.Ta., A.-.L.P., W.C.B. Writing—editing: all authors. Results interpretation: all authors. Kara Britt did the laboratory experiment—estrogen pellet implantation.

## Competing interests

W.C.B. is a co-inventor on 'Methods of treating neurodevelopmental diseases and disorders', USA Patent No. US9925163B2, Australian Patent No. 2015271652. This has been licensed to Meizon Innovation Holdings. A.-L.P. is a scientific advisor and W.C.B. is a board member of the Meizon Innovation Holdings. The remaining authors declare no competing interests.

## Additional information

Christos Symeonides [1,2,3,29,17], Kristina Vacy [4,5,29,17], Sarah Thomson [4], Sam Tanner [4], Hui Kheng Chua [4,6], Shilpi Dixit [4], Toby Mansell [2,7], Martin O'Hely [2,8], Boris Novakovic [2,8], Julie B. Herbstman [9,10], Shuang Wang [9,11], Jia Guo [9,11], Jessalynn Chia [4], Nhi Thao Tran [4,28], Sang Eun Hwang [4], Kara Britt [12,13,14], Feng Chen [4], Tae Hwan Kim [4], Christopher A. Reid [4], Anthony El-Bitar [4], Gabriel B. Bernasochi [4,15], Lea M. Durham Delbridge [15], Vincent R. Harley [12,16], Yann W. Yap [6,16], Deborah Dewey [17], Chloe J. Love [8,18], David Burgner [2,7,19,20], Mimi L. K. Tang [2,15], Peter D. Sly [8,21,22], Richard Saffery [2], Jochen F. Mueller [23], Nicole Rinehart [24], Bruce Tonge [25], Peter Vuillermin [2,8,18], the BIS Investigator Group*, Anne-Louise Ponsonby [2,3,4,30,18] & Wah Chin Boon [4,26,30,18] ✉

[1]Minderoo Foundation, Perth, Australia. [2]Murdoch Children's Research Institute, Parkville, Australia. [3]Centre for Community Child Health, Royal Children's Hospital, Parkville, Australia. [4]The Florey Institute of Neuroscience and Mental Health, Parkville, Australia. [5]School of Population and Global Health, The University of Melbourne, Parkville, Australia. [6]The Hudson Institute of Medical Research, Clayton, Australia. [7]Department of Pediatrics, The University of Melbourne, Parkville, Australia. [8]School of Medicine, Deakin University, Geelong, Australia. [9]Columbia Center for Children's Environmental Health, Columbia University, New York, NY, USA. [10]Department of Environmental Health Sciences, Columbia University, New York, NY, USA. [11]Department of Biostatistics, Columbia University, New York, NY, USA. [12]Department of Anatomy and Developmental Biology, Monash University, Clayton, Australia. [13]Breast Cancer Risk and Prevention Laboratory, Peter MacCallum Cancer Centre, Melbourne, Australia. [14]Sir Peter MacCallum Department of Oncology, The University of Melbourne, Melbourne, Australia. [15]Faculty Medicine, Dentistry & Health Sciences, University of Melbourne, Parkville, Australia. [16]Sex Development Laboratory, Hudson Institute of Medical Research, Clayton, Australia. [17]Departments of Paediatrics and Community Health Sciences, The University of Calgary, Calgary, Canada. [18]Barwon Health, Geelong, Australia. [19]Department of General Medicine, Royal Children's Hospital, Parkville, Australia. [20]Department of Pediatrics, Monash University, Clayton, Australia. [21]Child Health Research Centre, The University of Queensland, Brisbane, Australia. [22]WHO Collaborating Centre for Children's Health and Environment, Brisbane, Australia. [23]Queensland Alliance for Environmental Health Sciences, The University of Queensland, Brisbane, Australia. [24]Monash Krongold Clinic, Faculty of Education, Monash University, Clayton, Australia. [25]Centre for Developmental Psychiatry and Psychology, Monash University, Clayton, Australia. [26]School of BioSciences, Faculty of Science, The University of Melbourne, Parkville, Australia. [28]Present address: The Ritchie Centre, Department of Obstetrics and Gynaecology, School of Clinical Sciences, Monash University, Clayton, Australia. [29]These authors contributed equally: Christos Symeonides, Kristina Vacy. [30]These authors jointly supervised this work: Anne-Louise Ponsonby, Wah Chin Boon. *A list of authors and their affiliations appears at the end of the paper. ✉e-mail: wah.chin.boon@florey.edu.au

## the BIS Investigator Group

Christos Symeonides [1,2,3,29,17], Toby Mansell [2,7], Martin O'Hely [2,8], David Burgner [2,7,19,20], Mimi L. K. Tang [2,15], Peter D. Sly [8,21,22], Richard Saffery [2], Jochen F. Mueller [23], Peter Vuillermin [2,8,18], Fiona Collier [2,8,18], Anne-Louise Ponsonby [2,4], Leonard C. Harrison [15,27], Sarath Ranganathan [2,3,15], & Lawrence Gray [2,8,18]

[27]Walter and Eliza Hall Institute, Parkville, Australia.

