## [Peer Review File · Nature Communications]

REVIEWER COMMENTS

Reviewer #1 (Remarks to the Author):

This was an ambitious study designed to test the hypotheses that mid-gestation exposure to BPA increases the risk for ASD symptoms in males by acting through methylation and reduced expression of CYP19A1, and that this deficiency may be reversed by postnatal administration of 10HDA by mimicking the effects of estrogen. While the study has potentially high impact and other strengths, including the integrated use of a human cohort, cell culture models, two mouse models, and genome-wide RNAseq of fetal BPA-exposed brain, the experiments themselves individually lack rigor, as described below.

Major concerns

1. In the human epidemiology studies, there are multiple concerns.

a. First, for ASD diagnosis analysis at age 4, there were only 9 doctor-diagnosed ASD within a cohort of 791. This is problematic for the very low number of ASD cases and the lack of a standard diagnostic instrument (ADOS/ADR-I).

b. The second analysis of “autism symptoms” was assessed by DSM-5-oriented autism spectrum problems (ASP) scale of the Child Behavior Checklist for Ages 1.5-5 (CBCL) administered at 2-3 years. While the authors cite that the CBCL “strongly predicted diagnosed autism by age 4” the prevalence of 58.7% of subjects above the median CBCL compared to 1.1% for ASD diagnosis is inconsistent with predictive accuracy. It is not at all clear if the 9 subjects diagnosed with ASD actually have the highest CBCL scores or if T-scores above 50 reflect demonstrated differences in ASD symptoms in those not formally diagnosed with ASD.

c. The statistical models for assessing the significance of BPA exposure and ASD diagnosis or symptoms are nonstandard and problematic. A table showing all of the additional demographic and technical variables available and the p values of their association with ASD, CBCL, and BPA categories used to test the hypothesized associations. The choice of confounding variables to be included in the models needs to be determined systematically.

d. The sentence on p. 5 “Of the 676 infants with CBCL data in the cohort sample, 249 (36.8%) had an ASP score above median based on CBCL normative data (Table S1)” is not consistent with the data in Table S1 (58.7%).

e. The handling of genetic association was not acceptable, the genetic risk should be calculated by a weighted risk score for CYP19A1 and compared to polygenic risk score for ASD.

f. Table S4 is not an acceptable way to report on adjustment for confounding variables.

g. The human DNA methylation studies from cord blood would benefit from a comparison to CYP19A1 in other published ASD cord blood and postmortem brain studies. While BPA exposures in these studies are not known, high BPA exposure is common enough that hypermethylation of CYP19A should have been

seen in other studies. Alternatively, evidence for hypermethylation of Cyp19A brain promoter methylation in mouse would be an important replication of this ascribed mechanism.

2. The social behavioral test shown in Fig 4 and multiple places later lacks controls. Specifically, time interacting with stranger mouse needs to be presented as relative to the time spent interacting with the empty cage.

3. The bioinformatic analyses of RNAseq data lack rigor and the ability to adjust for biological and technical variables. The enrichment analyses in the analyses of RNAseq data need to have an appropriate comparison group to be meaningful, but that was not clear from the methods. Was CYP19A1 confirmed to be significantly downregulated by RNAseq? The list of DEGs with summary stats should be provided as a supplementary table.

4. For the BPA detection QC data, the level of detection (LOD) and the % of samples with levels above the LOD should be reported, as well as how samples falling below the LOD were assessed.

Minor concerns

1. The cell line “SH-S5Y5” should be “SH-SY5Y”.

2. Bar graphs should show individual values as dotplots.

3. It was not clear in the microscopy measurements were performed blinded to experimental treatments.

4. The statistical test used should be provided in every legend and text statement that includes a p value.

Reviewer #2 (Remarks to the Author):

The manuscript building relationship between prenatal BPA exposure, ASD and aromatase gene by human cohort studies. Based on the clinical findings, the authors then performed further laboratory studies to elucidate mechanism. The work close to the clinic research, which is beneficial to pregnant women to avoid pregnancy risks. I would recommend accepting it in Nature Communications after minor revisions. My comments are listed as below:

L.102 – In the human studies, please referenced literatures about the primary routes of exposure to hazardous BPA, especially for pregnant mothers.

In the supplement Table 4, should be exclude the negative impact of BPA in the breast milk for breast-fed babies from birth to 4 years age.

Are the alcohol, smoking ...exclusions self-reported and can those be trusted? e.g. could there be many more who have indulged during the trial?

Supplementary page 11, for whole genome SNP arrays, a brief method to harvest child whole cord blood should be mentioned.

BPA exposure reduce aromatase in SH-S5Y5 cells and Cyp19-EGFP mice, the reference with the treatment dose should be cited. In the supplement, "BPA injection administration treatment" section, at a dosage of 50 µg/kg (deemed as the safe consumption level by the Food and Drug Administration, FDA), please provide the website link to find information. Similarly, the reasoning behind the treatment dose of 10HDA, the corresponding literature should be cited.

L.221 – Please include the graph number corresponding to the result.

L.742 –for the Golgi staining and neuronal tracing experiment, ideally a visible neuronal image in the figures, if possible, please provide a representative imaging picture.

L.282-291–check the spelling of ER α and ER β . Please include the PDB number of crystal structure of estrogen receptors α (Er α) and β (Er β), as well as in the Computational Molecular Docking method in supplement.

L.350 –Prenatal BPA exposure promotes ASD at age 4 and 2, especially at age 2, the adverse effect of BPA on ASP was dramatically evident among males with a low aromatase enzyme activity and a low CYP19A1 genetic score. Please demonstrate the possible causes behind based on previous researches.

-Check the spelling, e.g. CO₂, add a space after the unit.

-It is not clear what the label in the figures –such as the unit "µg/L", "µg/kg"

-Authors need to clearly state the instruments and reagents used for experiment.

- Authors need to clearly specify the number of animals used in each oral administration or injection treatment, and need to specify the volume of injection.

-Why authors didn't complete the mechanism finding based on RNAseq data, unless these data in another work.

Reviewer #3 (Remarks to the Author):

This is a very interesting study and there are several strengths. However, there are some major concerns that need addressing.

Major strengths

1. Novel hypothesis.
2. Comprehensive experimental design with use of in vitro and in vivo models and multiple behavioral endpoints.
3. Use of the Aromatase KO and the Aromatase-GFP models
4. Robust controls
5. In depth statistical analyses.

Major concerns.

1. The dose used for the in vitro model is quite high. It is also not clear why a lower dose of BPA was not included in the in vivo model. Please discuss these limitations in the results or Discussion. Further, BPA levels in the in vivo studies were not included which is critical to determine the relevance of the model to human exposures.
2. The human neuroblastoma cell line (SH-SY5Y) is not a relevant cell line for these studies. Primary cell culture of neonatal neurons is highly recommended.
3. Bulk RNAseq was performed, but no single-cell RNAseq analyses were included. This is problematic in the brain as there are multiple cell-types with multiple cell-state (activated neurons, etc.). Differences in the transcriptome may reflect changes in cell type and cell-state. While it is out of the scope of the current study to add scRNAseq experiments, a discussion of the limitations must be included. The most important aspect of this study is the phenotype. The descriptive RNAseq experiments only marginally

add to the overall story. Line 329 states that BPA downregulated genes related to dendrite extension, yet the S7 figure legend states that these changes do not persist when adjusted for multiple testing. This is problematic and should not be buried in a figure legend in supplemental data. Please remove any sentences stating there are differences in gene expression unless those changes have been adjusted for multiple testing or measured by a different method such as qPCR. In addition, it would be helpful to include how many genes overall had differences in expression at a q-value <0.05 .

4. Genome-wide DNA methylation studies of cord blood and brain tissue is problematic as there are multiple cell types with multiple cell states (activated macrophages, t-cells, etc.). The reported changes were very small and differences in DNA methylation are highly likely to reflect changes in cell type and cell state. It also appears that there were no corrections for multiple testing. Finally, what about the rest of the data? Epic arrays were performed, but there was no discussion of the results other than CYP19A1 and Cyp19a1. I suggest eliminating these experiments from the paper.

Response to reviewers re NCOMMS-22-49038A

Prenatal programming of autism spectrum disorder by BPA occurs through aromatase disruption and is recovered by 10HDA therapy.

Thank you for these helpful reviewer comments and suggestions which we have responded to as outlined below. In particular, we are pleased to report that we have replaced the outcome of autism spectrum disorder at age 4 (9 verified cases) with autism spectrum disorder at age 9 years (n=43 verified cases). In addition, the later cases have also been verified now against DSM-V criteria. We are also pleased to report that we were able to bring in a second human cohort, the Columbia Centre for Children's Health Mothers and Newborns (CCCEH-MN) study that published a seminal paper on how higher prenatal bisphenol A levels were associated with hypermethylation of Brain-Derived Neurotrophic Factor (BDNF) in the male cord blood of infants in that birth cohort in the Proceedings of the National Academy of Science in 2015, a paper with over 200 citations to date. Not only were we able to replicate their 2015 finding, but they replicated our finding in the BIS cohort that higher prenatal BPA levels were associated with hypermethylation of the aromatase gene. Further, in both birth cohorts, using the modern causal inference technique of molecular mediation, we have demonstrated that the adverse effect of higher BPA on BDNF hypermethylation, shown previously to relate to reduce BDNF expression, is mediated through methylation of the aromatase gene, a key gene in sexually- dimorphic brain development.

We have conducted new work which we have included as new sections in the results.

They are:

- BPA effects on ASD diagnosis at 9 years are most evident in boys with a low CYP19A1 genetic score for aromatase enzyme activity. Pg. 7
- A sensitivity analysis of an additional weighted genetic function score, contained in the section "BPA effects on ASD symptoms at age 2 years are most evident in boys with a low CYP19A1 genetic score for aromatase enzyme activity". Pg. 6-7
- Replication of the association between higher BPA levels and hypermethylation of the aromatase brain promoter. Pg. 8
- Molecular mediation of higher BPA levels and hypermethylation of BDNF through higher methylation of aromatase. Pg. 8

We have also included extra figures and supplementary files:

- The addition of representative Golgi images to Figure 6: **(C)** Representative photomicrographs of Golgi staining of layer V of the somatosensory cortex of vehicle and BPA exposed mice. Pg. 32
- Supplementary Fig. 5 | NeuN staining of the Medial Amygdala Supplementary Pg. 20
- Supplementary file: Video demonstrating Molecular docking of E2 and 10HDA with estrogen receptor β Fig 7, Pg. 34
- Supplementary file: Video demonstrating Molecular docking of E2 and with estrogen receptor β Fig 7, Pg. 34
- Supplementary file: Tab1: maleBPA_10HDA_DEGs .xlsx: Differentially Expressed Genes in cortical primary cell culture from BPA exposed mice, male BPA exposed vs male control. Tab 2: RNASeq list of Differentially Expressed Genes in cortical primary cell culture treated with 10HDA for male 10HDA vs male control.

REVIEWER COMMENTS

Reviewer #1 (Remarks to the Author):

This was an ambitious study designed to test the hypotheses that mid-gestation exposure to BPA increases the risk for ASD symptoms in males by acting through methylation and reduced expression of CYP19A1, and that this deficiency may be reversed by postnatal administration of 10HDA by mimicking the effects of estrogen. While the study has potentially high impact and other strengths, including the integrated use of a human cohort, cell culture models, two mouse models, and genome-wide RNAseq of fetal BPA-exposed brain, the experiments themselves individually lack rigor, as described below.

Major concerns

1. In the human epidemiology studies, there are multiple concerns.

a. First, for ASD diagnosis analysis at age 4, there were only 9 doctor-diagnosed ASD within a cohort of 791. This is problematic for the very low number of ASD cases and the lack of a standard diagnostic instrument (ADOS/ADR-I).

We have now revised the paper to remove the ASD diagnosis by age 4 and replaced these with ASD cases by age 9 (n=43 cases). These were verified using the DSM-V criteria by paediatric medical review.

See main text results Pg. 6, line 117-120.

See supplement, Pg. 2, line 43-31

b. The second analysis of “autism symptoms” was assessed by DSM-5-oriented autism spectrum problems (ASP) scale of the Child Behavior Checklist for Ages 1.5-5 (CBCL) administered at 2-3 years. While the authors cite that the CBCL “strongly predicted diagnosed autism by age 4” the prevalence of 58.7% of subjects above the median CBCL compared to 1.1% for ASD diagnosis is inconsistent with predictive accuracy. It is not at all clear if the 9 subjects diagnosed with ASD actually have the highest CBCL scores or if T-scores above 50 reflect demonstrated differences in ASD symptoms in those not formally diagnosed with ASD.

We cited a receiver operating characteristic (ROC) curve analysis that showed the utility of the CBCL ASP scale at 2 years for predicting ASD diagnoses at 4 years. An ROC curve analysis uses a continuous predictor variable and a dichotomous outcome variable. We've reworded the section, as below, to make this clearer. We've also added mean CBCL ASP scores for ASD cases vs non-cases to show that ASD cases do in fact have higher scores compared to non-cases. In the analyses in the paper, we used above median CBCL ASP scores based on normative data (36.8% in our sample) vs below as a binary outcome variable. This cut-point has 100% sensitivity and 62.6% specificity for predicting ASD cases and provided high enough numbers to conduct the intended stratified analyses.

“In BIS, the DSM-5 oriented autism spectrum problems (ASP) scale of the Child Behavior Checklist (CBCL) at age 2 years³¹ predicted diagnosed autism strongly at age 4 and moderately at age 9 in receiver operating characteristic (ROC) curve analyses; area under the curve (AuC) of 0.92 (95% CI 0.82, 1.00)³² and 0.70 (95% CI 0.60, 0.80), respectively. The median CBCL ASP score in ASD cases and non-cases at 9 years was 50 (IQR=50, 51) and 51 (IQR=50, 58) respectively. Only ASD cases with a pediatrician-confirmed diagnosis of ASD against the DSM-V, as verified by the 30th of June 2023, were included in this report (see Supplementary Information).”

(Pg.6, line 121-128)

c. The statistical models for assessing the significance of BPA exposure and ASD diagnosis or symptoms are nonstandard and problematic. A table showing all of the additional demographic and technical variables available and the p values of their association with ASD, CBCL, and BPA categories used to test the hypothesized associations. The choice of confounding variables to be included in the models needs to be determined systematically.

We have reconducted the prospective analysis of the BPA-ASD association at ages 2 and 9 years using a matched case-control approach so as to allow the associations to be conditional as is standard in epidemiological cohort measurements of toxicants and subsequent human outcomes¹. We mainly focus on the BPA-ASD association with low genetic aromatase activity function score subgroup, where strong BPA findings are observed. However, the other technical and demographic factors have weaker associations that cannot be fully evaluated in this subgroup. We do provide new numerical readouts for table 4S, not merely a summary, that clearly demonstrate the effect of additional potential confounders of the BPA-ASD at age 2 association, where numbers did suffice (n=80). We have used a systematic approach and published our pipeline in the International Journal of Epidemiology in 2021. We now also point out in the footnote of this table how this encompasses epidemiological recommendations (e.g. potential confounders should change the estimate by more than 15% to be included¹)

Supplementary table S4, Pg. 85

d. The sentence on p. 5 “Of the 676 infants with CBCL data in the cohort sample, 249 (36.8%) had an ASP score above median based on CBCL normative data (Table S1)” is not consistent with the data in Table S1 (58.7%).

The numbers reported in the text of the paper were correct. The numbers in Supplementary Table 1 have now been corrected.

e. The handling of genetic association was not acceptable, the genetic risk should be calculated by a weighted risk score for CYP19A1 and compared to polygenic risk score for ASD.

We have now replicated our findings using an independent weighted genetic score for aromatase gene expression constructed from expression quantitative trait loci (eQTLs) available through the GTEx database. All of the SNPs used to create the additional score are new. It is good that the same pattern is found for both genetic scores. These can be found in Supplementary Table 3.

“Further, we replicated past work that higher prenatal BPA levels are associated with BDNF hypermethylation², previously demonstrated to be associated with lower BDNF expression in males².

Pg.15 line 425-427

f. Table S4 is not an acceptable way to report on adjustment for confounding variables.

Table S4 has been reworked as described in response 1d. We now provide numerical results in a standard format and reference a classic epidemiology reference book. See Supplementary Table 4.

g. The human DNA methylation studies from cord blood would benefit from a comparison to CYP19A1 in other published ASD cord blood and postmortem brain studies. While BPA exposures in

these studies are not known, high BPA exposure is common enough that hypermethylation of CYP19A should have been seen in other studies. Alternatively, evidence for hypermethylation of Cyp19A brain promoter methylation in mouse would be an important replication of this ascribed mechanism.

We have now included in the Discussion the findings from a meta-analysis of CYP19A1 promoter hypermethylation across two EWAS for ASD that are positively associated with ASD.

“Further, studying cord blood gene methylation as an outcome we have demonstrated that BPA exposure specifically methylated the offspring *CYP19A1* brain promoter in the BIS cohort, replicated in the CCCEH-MN cohort. Previously, in a meta-analysis of two epigenome-wide association studies (EWAS), two CpGs in the region around the brain promoters P1f and PII were significantly associated ($P < .05$) with ASD in both EWAS⁶⁰”

Pg. 15, line 423-427.

We also now replicate the higher BPA-Cyp19A1 finding in the BIS cohort with the CCCEH cohort and provide evidence in both that the effect of higher BPA on higher BDNF methylation is mediated through aromatase promoter methylation.

“In BIS, we aimed to reproduce these BDNF findings and extend them to investigate Aromatase methylation as a potential mediator of the BPA-BDNF relationship. This is a plausible mechanism given that the Aromatase/ER alpha pathway is involved in the activation of CREB, a key BDNF regulator (ref). In BIS, male infants exposed to the top quartile of BPA had greater methylation of the BDNF CREB-binding site (adjusted mean increase = 0.0027, $p = 0.02$). This was also evident overall (adjusted mean increase = 0.0023, $p = 0.006$) but not for females alone (adjusted mean increase = 0.0019, $p = 0.13$). We then assessed whether methylation of Aromatase promoter P1.f mediates this association. In both cohorts, aromatase methylation was positively associated with BDNF CREB-binding-site methylation in males (BIS, adjusted mean increase = 0.07, $p = 0.0008$; CCCEH, adjusted mean increase = 1.29, $p = 1.04 \times 10^{-11}$). Further, there was evidence that the adverse effect of BPA on BDNF methylation was mediated partly through aromatase in both cohorts (BIS, indirect effect, $p = 0.012$; CCCEH, indirect effect, $p = 0.012$).”

Pg. 8, line 201-213

1.2. The social behavioral test shown in Fig 4 and multiple places later lacks controls. Specifically, time interacting with stranger mouse needs to be presented as relative to the time spent interacting with the empty cage.

The only controls we evaluated were comparing ArKO to WT, BPA exposed to vehicle exposure and 10HDA treatment to vehicle treatment.

We did not provide the time spent interacting with the stranger relative to the time spent interacting with the empty cage (during trial 2 – social approach) because the test mouse was already exposed to the empty cage during the trial 1 – the habituation trial. The cage is not novel relative to stranger 1, time spent interacting there would be no more meaningful than time spent at the other non-interaction zones in the apparatus. Thus, a preference for the stranger over the empty cage doesn't necessarily rule out a deficit in social interaction.

We also controlled for potential side bias (the mouse preferring to interact with one particular cage over the other) during trial 2 by placing the stimulus mouse in the cage which was visited the least

by the test mouse during trial 1, the habituation trial. This is stated in Supplementary Information Pg. 8, line 308-313.

1.3.1. The bioinformatic analyses of RNAseq data lack rigor and the ability to adjust for biological and technical variables.

We now included the quality control steps into the extended methods (Supplementary Information Pg. 14, line 557-566) and revised the footnote of Supplementary Fig. 7. This now has a more detailed explanation.

“The libraries were sequenced with 50bp single end reads using an Illumina HiSeq and read quality assessed using FastQC (<http://www.bioinformatics.bbsrc.ac.uk/projects/fastqc/>). Untrimmed reads were aligned to mouse mm10 genome using Subjunc aligner (version 1.4.4) within the Subread package⁵. Sequencing data was then summarized into reads per transcript using Feature counts⁶ and the Gencode gene models for the mouse GRCm38/mm10 genome build (August 2014 freeze)⁷. Normalisation and statistical analysis on the count data was executed using EdgeR (version edgeR_3.4.2 in R studio, R version 3.0.2)⁸ after removing features with less than 10 counts per million for at least 3 of the samples. The data was scaled using trimmed mean of M-values (TMM)⁹ and differentially expressed genes between all treatment group (Benjamini–Hochberg false discovery rate >0.1). Annotation was added using the ensemble mouse gene annotation added using bioMart package¹⁰.”

Supplementary Pg. 14, line 557-566

1.3.2. The enrichment analyses in the analyses of RNAseq data need to have an appropriate comparison group to be meaningful, but that was not clear from the methods.

Of note the comparisons are not just between mouse groups (The comparison groups are BPA: male BPA exposure compared to male vehicle controls, 10HDA: male 10HDA exposure compared to male vehicle controls) but also between how BPA and 10HDA as treatments differentially effect the RNASeq-based pathways. The footnote below now details all these comparisons.

“Supplementary Fig. 7 | Shared brain-related pathways for 10HDA and BPA RNA-sequencing results.

(A) Gene Set Enrichment Analysis of RNA-sequencing data reveals BPA induces downregulation of processes related to intracellular respiration and metabolism compared to controls. Brain development has high energy demands, generating a need for optimal metabolic regulation¹¹, dysregulation of which has been associated with autism¹². These findings do not persist after FDR adjustment.

(B) Gene Set Enrichment Analysis of RNA sequencing data reveals 10HDA induces upregulation of brain, synaptic, and dendritic pathways compared to controls. Of these upregulated mechanisms, 413 persist after FDR correction. Using a candidate pathway approach, we observed that BPA downregulated, and 10HDA upregulated pathways ($n = 34$) with the keyword ‘axon’ ($n = 55$) or dendrite more than expected on chance (Fisher’s exact test, $P = 0.0144$ and $P = 0.0001$, respectively).

Biological processes in ASD pluripotent stem cell model. We then examined the top mid-gestation biological processes identified to be over-represented in autism compared to non-cases in a previous human pluripotent stem cell analysis (ref 57). Among the 11 (of 15) pathways also available in our analysis for both BPA and 10HDA, 82% ($n=9$) were altered in opposite directions for BPA and 10HDA, more than expected by chance (binomial test, $P = 0.033$)”

Supplementary Figure S7

1.3.3. Was CYP19A1 confirmed to be significantly downregulated by RNAseq?

Cyp19A1 was not in the library due to undetectable levels of aromatase expression in the fetal brain thus we could not report on its differential expression. Instead, we have provided pathway analysis of downstream targets of interest (figure 10), page 38.

Also mentioned in supplementary Pg. 14, line 540-542

3.4. The list of DEGs (differential expressed genes) with summary stats should be provided as a supplementary table.

An editorial summary table now provides the summary statistics for the DEGs in the supplementary files (see supplementary files).

1.4. For the BPA detection QC data, the level of detection (LOD) and the % of samples with levels above the LOD should be reported, as well as how samples falling below the LOD were assessed.

This is already reported in 'Other statistical analyses' of the Human Studies section in the supplement as well as in Supplementary Table 2. To improve clarity, we now refer readers of the main text to this information.

"Quality control information for the measurement of BPA is presented in Supplementary Table 2."
Pg. 6, line 130-131.

Minor concerns

Minor 1. The cell line "SH-S5Y5" should be "SH-SY5Y".

Thank you. Corrected (Figure legend for fig 3). Main and supplementary documents have also been checked.

Minor 2. Bar graphs should show individual values as dot plots.

Thank you. Bar graphs now include individual datapoints.

Minor 3. It was not clear in the microscopy measurements were performed blinded to experimental treatments.

We have clarified that microscopy measurements were performed blinded to treatment in the methods:

"Stained slides were coded to ensure that morphological analysis was conducted by an observer who was blind to the animals' experimental group".

(Supplementary Pg. 9 line 342)

Identification and delineation of the left PD MeA throughout the various sections for each brain was completed blind to experimental group until regions for all brain specimens had been analyzed (Supplementary Pg. 12, line 466)

Stereological analysis - The experimenter was blind to experimental group during stereological analysis.

(Supplementary Pg. 12., line 484)

Minor 4. The statistical test used should be provided in every legend and text statement that includes a p value.

The statistical test has now been provided in every figure legend. They have not been added to the text statements to keep the text more readable, like other Nature Communications articles.

Reviewer #2 (Remarks to the Author):

The manuscript building relationship between prenatal BPA exposure, ASD and aromatase gene by human cohort studies. Based on the clinical findings, the authors then performed further laboratory studies to elucidate mechanism. The work close to the clinic research, which is beneficial to pregnant women to avoid pregnancy risks. I would recommend accepting it in Nature Communications after minor revisions. My comments are listed as below:

L.102 – In the human studies, please referenced literatures about the primary routes of exposure to hazardous BPA, especially for pregnant mothers.

We have now added to the discussion:

“Despite bans on its use in all infant products by the European Union in 2011 and the U.S. Food and Drug Administration in 2012, BPA remains widespread in the environment⁶⁶. The main source of human exposure to BPA is dietary contamination⁶⁷. Bisphenols are used in the production of common food contact materials, and migrate from those materials during use⁶⁸, including polycarbonate food and beverage containers and the epoxy linings of metal food cans, jar lids, and residential drinking water storage tanks and supply systems⁶⁹. Additional sources of exposure include BPA-based dental composites and sealants epoxies, as well as thermal receipts⁶⁹.”

Pg. 17 line 555-561.

*In the supplement Table 4, should be exclude the negative impact of BPA in the breast milk for breast-fed babies from birth to 4 years age.
Are the alcohol, smoking ...exclusions self-reported and can those be trusted? e.g. could there be many more who have indulged during the trial?*

We now state at the bottom of Supplementary Table 4 that maternal smoking habit questions were previously validated by urinary cotinine assay²⁰. Breastfeeding has been removed from Supplementary table 4.

Supplementary page 11, for whole genome SNP arrays, a brief method to harvest child whole cord blood should be mentioned.

We have included information on how the cord blood was collected.

“Blood from the umbilical cord was gathered at birth and then transferred into serum coagulation tubes (BD Vacutainer). Following this, the serum was separated using centrifugation as described elsewhere²¹”.

Supplementary Pg. 2, line 59-60

BPA exposure reduce aromatase in SH-S5Y5 cells and Cyp19-EGFP mice, the reference with the treatment dose should be cited. In the supplement, “BPA injection administration treatment” section, at a dosage of 50 µg/kg (deemed as the safe consumption level by the Food and Drug Administration, FDA), please provide the website link to find information. Similarly, the reasoning behind the treatment dose of 10HDA, the corresponding literature should be cited.

A reference for the FDA safe level for BPA has been added to the paper and the supplement in the sections shown below.

“This dose matches current USA recommendations^{38,39} as well as the Tolerable Daily Intake (TDI) set by the European Food Safety Authority (EFSA) at the time that the mothers in our human cohort were pregnant^{30,40}”

Pg. 9, line 232-235

To study the effects of prenatal BPA exposure on brain development, pregnant dams were subject to BPA at a dose of 50 µg/kg/day via subcutaneous injection, or a vehicle injection during a mid-gestation window of E10.5 to E14.5, which coincides with amygdala development.

Pg. 9, line 230-232

“Plugged FVBN dams were randomly assigned to receive daily scruff subcutaneous injections (24G x 1”, Terumo, Somerset, New Jersey, USA) of BPA (239658-50G, Sigma-Aldrich Co., St Louis, MO, USA) in ethanol and peanut oil (Coles, Victoria, Australia), either between E0.5-E9.5, E10.5-E14.5 or E15.5-birth at a dosage of 50 µg/kg (deemed as the safe consumption level by the Food and Drug Administration, FDA)²³ or 0 µg/kg (vehicle) of maternal body weight”

Supplement, Pg. 7 line 264-270

L.221 – Please include the graph number corresponding to the result.

The figure corresponding to this result was not originally included. We have now added it to the supplementary (Supplementary Figure 5) and included the figure number into the main text.

L.742 –for the Golgi staining and neuronal tracing experiment, ideally a visible neuronal image in the figures, if possible, please provide a representative imaging picture.

We have added some representative Golgi images of the cortex of BPA exposed mice compared to controls into Figure 6. Furthermore, we also included some images of the neural tracing in the extended methods in the supplementary

Supplementary Pg. 9 and 10

L.282-291—check the spelling of ER α and ER β . Please include the PDB number of crystal structure of estrogen receptors α (Er α) and β (Er β), as well as in the Computational Molecular Docking method in supplement.

The PDB numbers have now been included in main text, figure caption and in the extended methods.

1. Main text – page 12, line 349
2. Fig – page 35. Figure 7
3. Sup. - Page 15 line 587

L.350 –Prenatal BPA exposure promotes ASD at age 4 and 2, especially at age 2, the adverse effect of BPA on ASP was dramatically evident among males with a low aromatase enzyme activity and a low CYP19A1 genetic score. Please demonstrate the possible causes behind based on previous researches.

We now report this finding for ASD cases at 9 years now, a larger case group with DSM-V verification (Pg. 6, line 116-128).

The Discussion outlines how past findings from other research are consistent with a causal interpretation of the findings reported here.

In summary, our data suggest that a lack of aromatase leads to ASD like chamber in mice, as does BPA exposure in utero. BPA exposure also leads to a downregulation of aromatase expression and related pathways. Thus, boys with low aromatase are more vulnerable to the endocrine disrupting effects of BPA. We have referenced relevant work throughout to support this, but this exact issue has not been previously studied.

-Check the spelling, e.g. CO₂, add a space after the unit.

We have changed some instances of CO₂ to CO₂. Ug changed to μ g

-It is not clear what the label in the figures –such as the unit “ μ g/L”, “ μ g/kg”

The font of letter ‘ μ ’ has been changed from ‘u’ to ‘ μ ’ for increased clarity on all the figures

-Authors need to clearly state the instruments and reagents used for experiment.

We have added additional detailed information on the instruments and reagents in the supplementary where it was missing, e.g., for 10HDA (e.g., supplementary Pg.14, line 552).

- Authors need to clearly specify the number of animals used in each oral administration or injection treatment and need to specify the volume of injection.

We have added the sample size information in the figure captions where they were missing

throughout.

Injection volume has been added to the supplementary methods, supplementary Pg. 7 line 269 and 276.

Reviewer #3 (Remarks to the Author):

This is a very interesting study and there are several strengths. However, there are some major concerns that need addressing.

Major strengths

- 1. Novel hypothesis.*
- 2. Comprehensive experimental design with use of in vitro and in vivo models and multiple behavioral endpoints.*
- 3. Use of the Aromatase KO and the Aromatase-GFP models*
- 4. Robust controls*
- 5. In depth statistical analyses.*

Major concerns.

- 1. The dose used for the in vitro model is quite high. It is also not clear why a lower dose of BPA was not included in the in vivo model. Please discuss these limitations in the results or Discussion. Further, BPA levels in the in vivo studies were not included which is critical to determine the relevance of the model to human exposure.*

The human and mouse work were done in parallel; the in vivo model's dose was chosen based on the FDA allowable exposure levels. The lower levels in the human work only became available at the final analysis stage. However, the dose a mouse dose still corresponds to the top 0.5% of our cohort, and other global populations may experience higher exposures to BPA.

We have added additional information in the Discussion, Pg. 18-19, line 525-555.

“Notably, while we find adverse association at 13mg/kg, we do not have sufficient participants with lower exposure to evaluate a safe lower limit of exposure below this. Our findings in cell culture, with concentration 5 µg/L, parallels these human findings in terms of dose response. Although there are limitations in translating concentrations across body compartments without a stronger understanding of pharmacokinetics of BPA, 5 µg/L corresponds to the 90th and 95th percentile of BPA in urine in our human cohort and allowing for a standard factor of 10 for variability in human sensitivity used when setting TDIs [58], indicates relevance down to at least 0.5 µg/L, below the median urine concentration. Our findings in laboratory animal studies, with exposure of 50 µg/kg bodyweight, are a little higher, as they were designed to correspond to the current RfD and TDI, but implications nevertheless have relevance within the range of exposure in our cohort. Allowing for standard factors of 10 for interspecies variability [58] and variability in human sensitivity [58], our animal study findings support a TDI at 500 ng/kg or below, which corresponds to the upper 0.5% of our human cohort. The findings of the human study, also allowing for a factor of 10 for variability in human sensitivity⁶³, our results, therefore, support a more conservative RfD/TDI at or below 1.3 ng/kg⁶². “

Pg. 18-19, line 540-554.

2. The human neuroblastoma cell line (SH-SY5Y) is not a relevant cell line for these studies. Primary cell culture of neonatal neurons is highly recommended.

SH-SY5Y were originally chosen because they were known to express aromatase and SH-SY5Y have been used in BPA and ASD research²⁶

Supplementary Pg.5 line 185-187

As suggested by a reviewer, we repeated the human neuroblastoma cell line (SH-SY5Y) study on BPA's effects on aromatase expression using the primary mouse neuronal cell culture derived from E14.5 fetal mouse brain cortices. However, levels of aromatase expression in primary mouse neuronal cell culture are below the detection limit of Western Blot or qPCR. This is not unexpected because only a few cells in the fetal brain expressed aromatase as shown in our immunohistological staining Supplementary Fig. 1. This is consistent in that Aromatase+ cells represent <0.05% of neurons in the adult mouse brain²⁷.

“Because of undetectable levels of cyp19 RNA in the fetal brain, cyp19 RNA levels were not included. This is consistent in that Aromatase+ cells represent <0.05% of neurons in the adult mouse brain.”

Supplementary Pg. 14, line 538-542

Nonetheless, we demonstrated that the aromatase+ neuron population in the medial amygdala decreased ($P=0.004$) in male BPA exposed mice compared to control mice (Pg. 29, Fig 3.)

3. Bulk RNAseq was performed, but no single-cell RNAseq analyses were included. This is problematic in the brain as there are multiple cell-types with multiple cell-state (activated neurons, etc.). Differences in the transcriptome may reflect changes in cell type and cell-state. While it is out of the scope of the current study to add scRNAseq experiments, a discussion of the limitations must be included.

The following sentence is added to the Discussion:

“The changes we observed in our RNAseq data could be due changes in the cell type or cell state in this study. This could be clarified by future single cell RNAseq experiments now that this specific issue has been identified.”

Pg. 18 line 521-523,

The most important aspect of this study is the phenotype. The descriptive RNAseq experiments only marginally add to the overall story. Line 329 states that BPA downregulated genes related to dendrite extension, yet the S7 figure legend states that these changes do not persist when adjusted for multiple testing.

This is problematic and should not be buried in a figure legend in supplemental data.

The analysis of the Gene Ontology “dendrite extension” pathway reported in the main text relates to figure S6 and was motivated by our experimental findings re: reduced dendrite length under BPA exposure and increased length under 10HDA administration. This was a candidate-pathway approach and so correction for multiple comparisons was not required. The results reported in

figure S7 (supplementary Pg. 22), where the issue of multiple comparisons is noted, relate to the broader agnostic test across all available Gene Ontology categories. Here, too, however, while no individual pathways remained significant for BPA after adjustment, we found a significant BPA-associated down-regulation, and 10HDA-associated upregulation, of pathways with the term “axon” and “dendrite”, as we report (line 355-356, figure S7 caption). Like the finding in figure S6, this hypothesis-driven result involved a single test for each keyword (Fisher’s exact test), and so did not require p-value correction.

We have updated the results in the main text to highlight this:

“Firstly, pathway analysis of the RNAseq data was performed for Gene Ontology (GO) categories using the clusterProfileR R package (refer to Supplementary Information). No individual pathways in the BPA analysis survived correction for multiple comparisons using an agnostic (non-candidate) approach. Further candidate investigation using the binomial test showed a significant inverse effect of BPA and 10HDA on pathways previously linked to autism⁵⁹, with 10HDA treatment counteracting the effects of BPA on these pathways (Supplementary Fig. 7). Based on our Golgi staining experimental findings relating to altered dendrite morphology, we further assessed the category “dendrite extension” as a candidate pathway. Genes in this pathway were downregulated by BPA (Supplementary Fig. 8A) and upregulated by 10HDA (Supplementary Fig. 8B). More broadly, Fisher’s exact test showed a significant BPA-associated down-regulation ($P=0.01$), and 10HDA-associated up-regulation ($P=0.0001$), of pathways with the terms “axon” and “dendrite”. Notably, the majority (82%; 9 of the top 11 available) mid gestational biological processes whose activity is overrepresented in induced pluripotent stem cells of autism cases vs. controls⁵⁹ were impacted by BPA, and in the opposite direction to 10HDA ($P=0.03$) (Supplementary Fig. 7).

Next, we performed pathway enrichment analysis, also using a candidate pathway approach, of the RNAseq data using Ingenuity (Fig 10). Strikingly, the effects of BPA and 10HDA on gene expression were diametrically opposed across many functional domains (Fig. 10). For example, the canonical pathways “Synaptogenesis Signaling pathway” and “CREB signaling” were downregulated by BPA but upregulated by 10HDA. Similarly, key brain functions, e.g., growth of neurites and neural development were down regulated by BPA and reciprocally upregulated by 10HDA (Fig. 10). Taken together, prenatal BPA exposure is detrimental to gene expression through a mechanism that may be ameliorated by postnatal 10HDA administration.”

Pg. 14 line 388-410

Please remove any sentences stating there are differences in gene expression unless those changes have been adjusted for multiple testing Binomial or measured by a different method such as qPCR. In addition, it would be helpful to include how many genes overall had differences in expression at a q-value <0.05.

We have now included the DEG tables for BPA exposure and 10HDA exposure, including q-values in the supplementary data files.

We now also provide reports based on candidate pathway approaches that do not require multiple comparison testing penalty. Please see response to reviewer 1.3.1. For example, the finding below is now reported in the main text, results section.

“Notably, the majority (82%; 9 of the top 11 available) mid gestational biological processes whose activity is overrepresented in induced pluripotent stem cells of autism cases vs controls⁵⁹ were impacted by BPA, and in the opposite direction to 10HDA ($P=0.03$) (Supplementary Fig. 7).”

Pg 14. line 399-402

4. Genome-wide DNA methylation studies of cord blood and brain tissue is problematic as there are multiple cell types with multiple cell states (activated macrophages, t-cells, etc.). The reported changes were very small and differences in DNA methylation are highly likely to reflect changes in cell type and cell state.

This is an important issue. However, we do not believe the observed hypermethylation of aromatase can be explained by differences in cell type proportions. First, we included cell-type proportion estimates as covariates in all regression models, thus accounting for variability in methylation due to cell type. Second, we found the strongest association with BPA exposure directly over the brain promoter, consistent with BPA acting causally on brain development via aromatase methylation, but an unlikely coincidence if our findings are driven by variability in cell type.

Third, methylation across the aromatase brain promoter correlates strongly between blood and brain (Spearman rho = 0.94, $p < 2.2e-16$), meaning that brain-related changes can, indeed, be identified across this region in blood.

Pg. 8, line 186-187.

Fourth, as the updated manuscript indicates, we have now replicated this finding in an independent human cohort. This out of sample replication in a second birth cohort, the CCCEH-MN cohort, is an important addition to this manuscript.

Pg. 8, line 191-213

Fifth, we now report a meta-analysis of two human EWAS studies:

“Previously, in a meta-analysis of two epigenome-wide association studies (EWAS), two CpGs in the region around the brain promoters P1f and PII were significantly associated ($P < .05$) with ASD in both EWAS⁶⁰.”

Pg.15 line 425-427

It also appears that there were no corrections for multiple testing. Finally, what about the rest of the data? Epic arrays were performed, but there was no discussion of the results other than CYP19A1 and Cyp19a1. I suggest eliminating these experiments from the paper.

This was a candidate-gene approach to investigate whether the effects of BPA or a knockout of the aromatase gene observed in our mouse studies could be explained, mechanistically, by increased methylation of the aromatase promoter. Correction for multiple comparisons was not required as we were interested in differences in average methylation specifically across the promoter region of aromatase (i.e., we tested for an association between the BPA-exposure variable and a *single* variable summarising methylation across this region). We did not conduct an epigenome-wide association study across all CpGs available on the EPIC array. We have revisited the methods and results to ensure that this is clear throughout. Further, as stated above, we have strengthened this finding by providing a fourth and fifth piece of evidence, including replication in an out-of-sample human cohort, the US-based CCCEH-MN cohort.

References

1. Lash, T.L., VanderWeele, T.J., Haneuse, S. & Rothman, K.J. Modern Epidemiology. 1-1174 (Lippincott Williams & Wilkins, 2021).
2. Kundakovic, M., *et al.* DNA methylation of BDNF as a biomarker of early-life adversity. *Proc Natl Acad Sci U S A* **112**, 6807-6813 (2015).
3. Antaki, D., *et al.* A phenotypic spectrum of autism is attributable to the combined effects of rare variants, polygenic risk and sex. *Nature Genetics* **54**, 1284-1292 (2022).
4. Takahashi, N., *et al.* Association of Genetic Risks With Autism Spectrum Disorder and Early Neurodevelopmental Delays Among Children Without Intellectual Disability. *JAMA Netw Open* **3**, e1921644 (2020).
5. Liao, Y., Smyth, G.K. & Shi, W. The Subread aligner: fast, accurate and scalable read mapping by seed-and-vote. *Nucleic Acids Res* **41**, e108 (2013).
6. Liao, Y., Smyth, G.K. & Shi, W. featureCounts: an efficient general purpose program for assigning sequence reads to genomic features. *Bioinformatics* **30**, 923-930 (2013).
7. Harrow, J., *et al.* GENCODE: producing a reference annotation for ENCODE. *Genome Biol* **7 Suppl 1**, S4.1-9 (2006).
8. Robinson, M.D., McCarthy, D.J. & Smyth, G.K. edgeR: a Bioconductor package for differential expression analysis of digital gene expression data. *Bioinformatics* **26**, 139-140 (2010).
9. Robinson, M.D. & Oshlack, A. A scaling normalization method for differential expression analysis of RNA-seq data. *Genome Biol* **11**, R25 (2010).
10. Durinck, S., Spellman, P.T., Birney, E. & Huber, W. Mapping identifiers for the integration of genomic datasets with the R/Bioconductor package biomaRt. *Nat Protoc* **4**, 1184-1191 (2009).
11. Namba, T., Nardelli, J., Gressens, P. & Huttner, W.B. Metabolic regulation of neocortical expansion in development and evolution. *Neuron* **109**, 408-419 (2021).
12. Frye, R.E., *et al.* Prenatal air pollution influences neurodevelopment and behavior in autism spectrum disorder by modulating mitochondrial physiology. *Mol Psychiatry* **26**, 1561-1577 (2021).
13. DeRosa, B.A., *et al.* Convergent pathways in idiopathic autism revealed by time course transcriptomic analysis of patient-derived neurons. *Sci Rep* **8**, 8423 (2018).
14. Welch, C. & Mulligan, K. Does Bisphenol A Confer Risk of Neurodevelopmental Disorders? What We Have Learned from Developmental Neurotoxicity Studies in Animal Models. *Int J Mol Sci* **23**(2022).
15. Casas, M., *et al.* Dietary and sociodemographic determinants of bisphenol A urine concentrations in pregnant women and children. *Environ Int* **56**, 10-18 (2013).
16. Covaci, A., *et al.* Urinary BPA measurements in children and mothers from six European member states: Overall results and determinants of exposure. *Environ Res* **141**, 77-85 (2015).
17. Lewis, R.C., *et al.* Predictors of urinary bisphenol A and phthalate metabolite concentrations in Mexican children. *Chemosphere* **93**, 2390-2398 (2013).
18. Becher, R., *et al.* Presence and leaching of bisphenol a (BPA) from dental materials. *Acta Biomater Odontol Scand* **4**, 56-62 (2018).
19. Gerona, R.R., *et al.* Direct measurement of Bisphenol A (BPA), BPA glucuronide and BPA sulfate in a diverse and low-income population of pregnant women reveals high exposure, with potential implications for previous exposure estimates: a cross-sectional study. *Environ Health* **15**, 50 (2016).
20. Dwyer, T., Ponsonby, A.-L. & Couper, D. Tobacco Smoke Exposure at One Month of Age and Subsequent Risk of SIDS—A Prospective Study. *American Journal of Epidemiology* **149**, 593-602 (1999).

21. Burugupalli, S., *et al.* Ontogeny of circulating lipid metabolism in pregnancy and early childhood - a longitudinal population study. *Elife* **11**(2022).
22. EPA, U.S.E.P.A.U. Bisphenol A; CASRN 80-05-7. in *Integrated Risk Information System (IRIS) Chemical Assessment Summary* (National Center for Environmental Assessment, Washington, DC, USA, 1988).
23. U.S Food & Drug Administration. Bisphenol A (BPA): Use in Food Contact Application. (<https://www.fda.gov/food/food-additives-petitions/bisphenol-bpa-use-food-contact-application>, 2023).
24. Authority, E.F.S. Opinion of the Scientific Panel on food additives, flavourings, processing aids and materials in contact with food (AFC) related to 2, 2-BIS (4-HYDROXYPHENYL) PROPANE. *EFSA Journal* **5**, 428 (2007).
25. Vuillermin, P., *et al.* Cohort profile: The Barwon Infant Study. *Int J Epidemiol* **44**, 1148-1160 (2015).
26. Thongkorn, S., *et al.* Investigation of autism-related transcription factors underlying sex differences in the effects of bisphenol A on transcriptome profiles and synaptogenesis in the offspring hippocampus. *Biol Sex Differ* **14**, 8 (2023).
27. Unger, E.K., *et al.* Medial amygdalar aromatase neurons regulate aggression in both sexes. *Cell Rep* **10**, 453-462 (2015).

REVIEWER COMMENTS

Reviewer #1 (Remarks to the Author):

The authors have done a thorough job of trying to respond to prior reviewer critiques. I do think the manuscript is much improved. But I am still concerned that this representation of the social behavioral (3-chamber) task is non-standard and the manuscript should be further evaluated for soundness of the statistical methods.

Reviewer #3 (Remarks to the Author):

The authors have satisfactorily addressed my concerns

Reviewer #4 (Remarks to the Author):

Mouse social and repetitive behavior results are presented in Figure 4. The authors conducted an insightfully modified 3-chambered social approach task with great attention to methodological details. Unfortunately, insufficient data are shown.

1. Please display the number of seconds in which the nose of the subject mouse was pointed at the novel mouse (stranger) AND the number of seconds of nose point directed at the novel object (nylon mesh-covered wire cage), for each group.

The 3-chambered social approach assay is a comparison between interest in a target mouse versus interest in a target object. It was designed to model the tendency of children with autism to play with a favorite toy rather than playing with another child.

The reason that it is essential to show sniff time directed at both the novel object and the novel mouse is that the number of seconds spent with the novel mouse is influenced by general exploratory locomotion, which can be highly variable in mice. Motor, olfactory, visual, auditory, and other issues may

increase or decrease overall exploration, and therefore affect time spent with the novel mouse. Therefore the comparison to the novel object control group is required.

2. Statistical analysis of 3-chambered social approach data is conducted by comparing number of seconds sniffing the novel mouse versus number of seconds sniffing the novel object, WITHIN treatment group. Sociability is defined as more time with the novel mouse than with the novel object.

This assay is a binary Yes or No test. If significantly more time is spent with the novel mouse than novel object within a group, then that group displays sociability. If the comparison is not significant, then that group does not display sociability.

3-chambered social approach is not sensitive enough to compare absolute values of number of seconds with the novel mouse across genotypes, or across treatment groups. Again, high or low levels of general exploratory locomotion and sensory abilities within the 3-chambered apparatus are likely to introduce confounding artifacts.

We greatly appreciate the further feedback and suggestions. We note that there were two remaining issues requiring further revisions 1. the 3 chamber test descriptions and presentation of the findings and 2. statistical test checking. We now address each in turn. (1) We have now addressed comments on the 3 chamber test from reviewer 4 and (2) thoroughly checked our statistical analysis, including further review by two statisticians (see below).

Issue 1: The 3-chamber test. Reviewer comments and responses.

Reviewer #1: *But I am still concerned that this representation of the social behavioral (3-chamber) task is non-standard*

Reviewer #4: *Mouse social and repetitive behavior results are presented in Figure 4. The authors conducted an insightfully modified 3-chambered social approach task with great attention to methodological details. Unfortunately, insufficient data are shown.*

Response (General): We greatly appreciate your feedback that our modification is insightful. We have now made the further changes as recommended. That is:-

1. Originally and in the R1, Figure 4 shows social approach – that is differences between groups in time spent sniffing a novel conspecific “stranger” in the 3 chamber task. As requested, we have now focused on sociability analysis – that is comparing time sniffing a stranger to time spent sniffing an empty cage in the opposite chamber.
2. Figure 4 3-chamber data now include empty cage data.
3. To address the concern raised of insufficient data, we now include two further datasets from an additional experiment: The ArKO model early postnatal (PND5) estrogen replacement. We have updated the 3-chamber findings (Figure 4), and the stranger social interaction induced c-fos (Figure 5) to include these data. The additional experiment has brought in three further coauthors Nhi Thao Tran, Kara Britt and Sang Eun Hwang.

The following has been added to the manuscript:

Results Pg. 9, lines 269-275. The social preference towards stranger interaction zone compared to the empty zone was only evident for the wildtype ($P = 0.003$ Fig 4B) but not the ArKO ($P=0.45$ Fig 4B) This male specific social interaction deficit is similar to the BPA exposed pups. Further, post-natal 17β -estradiol (E2) replacement could reverse the ArKO reduction in sociability seen in males ($P = 0.04$, Supplementary Fig. 5) resulting in a similar stranger to empty preference in the E2 treated ArKO as observed for wildtype. The female ArKO pups did not have a sociability deficit (Supplementary Fig 5).

Results Pg. 11, lines 323-325. Similarly, we found that the MeA of ArKO mice had a marked deficit of 67% c-Fos-positive neurons when compared with WT ($P = 0.0002$; Fig. 5C) mice, which was ameliorated by early postnatal estradiol replacement.

Reviewer 4. 1. *Please display the number of seconds in which the nose of the subject mouse was pointed at the novel mouse (stranger) AND the number of seconds of nose point directed at the novel object (nylon mesh-covered wire cage), for each group.*

The 3-chambered social approach assay is a comparison between interest in a target mouse versus interest in a target object. It was designed to model the tendency of children with autism to play with a favorite toy rather than playing with another child.

The reason that it is essential to show sniff time directed at both the novel object and the novel mouse is that the number of seconds spent with the novel mouse is influenced by general exploratory locomotion, which can be highly variable in mice. Motor, olfactory, visual, auditory, and other issues may increase or decrease overall exploration, and therefore affect time spent with the novel mouse. Therefore the comparison to the novel object control group is required.

Response 4.1: Although we redesigned the 3-chamber apparatus itself, we followed the original the 3-chamber test paradigm whereby no novel object is introduced, only the stranger mouse¹. This allows comparison to previous work. For the future, we agree that the introduction of a novel object could be an elegant way to control for motor, olfactory, visual, auditory, and other issues. However, it is not possible for us to repeat all the 3-chamber studies.

We have now included the Y-maze test (supplementary data). Both the ArKO and BPA offspring did not show any differences with the respective control indicating that there is no motor, olfactory, visual, auditory, and other issues are unlikely in these animals .

The deficit appears social specific and unlikely to reflect general motor or sensory issues. This is because:

1. There are no differences between groups at the duration of time in the empty cage interaction zone in the three-chamber test
2. Concurrent y-maze data (which was conducted after the three-chamber test) revealed that there were no differences between groups. This is now reported in the results:

Results Pg. 10, lines 276-280

Further, we did not observe any behavioral differences between ArKO vs WT (or BPA exposed vs unexposed) mice and all groups in Y-maze test. All groups were able to distinguish the novel arm from the familiar arm and spent significantly more time in the novel arm compared to the familiar arm (data not shown), excluding major short-term memory, motor and sensory intergroup difference contributions.

Further, we have added in the discussion:

Discussion Pg. 15, lines 445-447

In the Y-maze test, both the ArKO and BPA offspring did not show any differences with the respective control indicating that there are no major memory, sensory or motor issues in these animals.

As recommended by the reviewer, we have now included the number of seconds the nose point was in the empty cage interaction zone, in addition to the number of seconds the nose point was directed at the stranger's cage. The empty cage is not a novel object, as the test mouse was habituated to it in the 10 minute trial prior. Trial 1 was a habituation trial, where there were two empty cages in the apparatus. Trial 2 was the introduction of the stranger into one of the cages, and the other was left empty. In addition, we now present this value as a portion of time spent at the zone rather than the absolute seconds to improve non specialised reader interpretation.

Reviewer 4. 2. *Statistical analysis of 3-chambered social approach data is conducted by comparing number of seconds sniffing the novel mouse versus number of seconds sniffing the novel object, WITHIN treatment group.*

This assay is a binary Yes or No test. If significantly more time is spent with the novel mouse than novel object within a group, then that group displays sociability. If the comparison is not significant, then that group does not display sociability.

3-chambered social approach is not sensitive enough to compare absolute values of number of seconds with the novel mouse across genotypes, or across treatment groups. Again, high or low levels of general exploratory locomotion and sensory abilities within the 3-chambered apparatus are likely to introduce confounding artifacts.

Response 4.2: We have now reanalysed the data using two way ANOVA, with cage type (stranger or empty) as the within groups factor (Fig 4). We have defined this as sociability in the figure legend of Figure 4.

Sociability is the higher proportion time spent in the stranger interaction zone compared to the empty interaction zone.

We considered confounding artifacts due to motor or sensory group differences to be unlikely for the reasons outlined in response 4.1

Issue 2: Statistical Review

Reviewer #1: The authors have done a thorough job of trying to respond to prior reviewer critiques. I do think the manuscript is much improved. The manuscript should be further evaluated for soundness of the statistical methods.

Dr Martin O’Hely, a mathematician, met with Kristina Vacy, Sam Tanner and Sarah Thomson and reviewed all their data analyses in the manuscript. Dr Martin O’Hely now satisfied criteria for authorship and has now been added to the author list. The Columbia CCC-MN cohort team also rechecked the findings from the US-based cohort.

In addition, an independent biostatistician, Alex Eisner, joined the review group for the statistical review process. He read and verified all statistical code. Alex Eisner has been added to the acknowledgement section. A number (<5) of minor issues emerged and were then corrected. These were minor revisions and did not materially alter any of the reported findings.

References

1. Rein B, Ma K, Yan Z. A standardized social preference protocol for measuring social deficits in mouse models of autism. *Nature Protocols* 2020; **15**(10): 3464-77.

REVIEWERS' COMMENTS

Reviewer #4 (Remarks to the Author):

The authors have comprehensively addressed the previous concerns. Sociability comparisons for the 3-chambered social approach test now include graphs showing time spent with the nose sniffing the novel mouse versus time spent with the nose sniffing the empty cage, within each treatment group. ANOVA statistics now evaluate the significance of sociability within group. Performance in a Y-maze test confirmed that the treatment groups did not differ on general exploratory locomotion. Additional experiments are included.

As recognized in the Rebuttal, when the cages are present during the habituation session, the choice during the sociability session becomes novel mouse versus familiar cage. For future experiments, the authors are advised to conduct the habituation session in an empty apparatus with no cages inserted.

Ns of 12-15 per group are encouraged for the future experimental design of mouse behavioral tests.

Reviewer Comments

Reviewer #4 (Remarks to the Author):

The authors have comprehensively addressed the previous concerns. Sociability comparisons for the 3-chambered social approach test now include graphs showing time spent with the nose sniffing the novel mouse versus time spent with the nose sniffing the empty cage, within each treatment group. ANOVA statistics now evaluate the significance of sociability within group. Performance in a Y-maze test confirmed that the treatment groups did not differ on general exploratory locomotion. Additional experiments are included.

As recognized in the Rebuttal, when the cages are present during the habituation session, the choice during the sociability session becomes novel mouse versus familiar cage. For future experiments, the authors are advised to conduct the habituation session in an empty apparatus with no cages inserted.

Ns of 12-15 per group are encouraged for the future experimental design of mouse behavioral tests.

Response: Thank you for this, we appreciate your comments.

In future studies, we will consider removing the empty cages during trial one and the sample size of 12-15 per group.

We used a smaller sample size for the ArKO mice test compared to the BPA exposed mice. This is because it is the dam which is exposed to the BPA, and there may be more variability between the pups in the amount of BPA they are exposed to. Thus in our sample size calculation, we must take into account the number of litters. This is not an issue for the ArKO mice, which are a genetic model.

We now also present the Y-maze data in the supplement.